# Soil microbial biomass, activity and community composition along altitudinal gradients in the High Arctic (Billefjorden, Svalbard)

Petr Kotas[1,2], Hana Šantrůčková[1], Josef Elster[3,4], Eva Kaštovská[1]

[1]Department of Ecosystem Biology, Faculty of Science, University of South Bohemia, České Budějovice, 370 05, Czech Republic

[2]Institute of Chemistry and Biochemistry, Faculty of Science, University of South Bohemia, České Budějovice, 370 05, Czech Republic

[3]Centre for Polar Ecology, Faculty of Science, University of South Bohemia, České Budějovice, 370 05, Czech Republic

[4]Centre for Phycology, Institute of Botany, Academy of Sciences of the Czech Republic, Třeboň, 379 82, Czech Republic

*Correspondence to:* Petr Kotas (kotyno@prf.jcu.cz)

**Abstract** The unique and fragile High Arctic ecosystems are vulnerable to global climate warming. The elucidation of factors driving microbial distribution and activity in arctic soils is essential for a comprehensive understanding of ecosystem functioning and its response to environmental change. The goals of this study were to investigate microbial biomass and activity, microbial community structure (MCS) and their environmental controls in soils along three elevational transects in coastal mountains of Billefjorden, Central Svalbard. Soils from four different altitudes (25, 275, 525, and 765 m above sea level) were analyzed for a suite of characteristics including temperature regimes, organic matter content, base cation availability, moisture, pH, potential respiration, and microbial biomass and community structure using phospholipid fatty acids (PLFA). We observed significant spatial heterogeneity of edaphic properties among transects, resulting in transect-specific effects of altitude on most soil parameters. We did not observe any clear elevation pattern in microbial biomass, and microbial activity revealed contrasting elevational patterns between transects. We found relatively large horizontal variability in MCS (i.e., between sites of corresponding elevation in different transects), mainly due to differences in the composition of bacterial PLFAs, but also a systematic altitudinal shift in MCS related to different habitat preferences of fungi and bacteria, which resulted in high fungi-to-bacteria ratios at the most elevated sites. The biological soil crusts on these most elevated, unvegetated sites can host microbial assemblages of a size and activity comparable to those of the arctic tundra ecosystem. The key environmental factors determining horizontal and vertical changes in soil microbial properties were soil pH, organic carbon content, soil moisture and $Mg^{2+}$ availability.

## 1 Introduction

Knowledge about the spatial distribution and activity patterns of soil microbial communities is essential to understand ecosystem functioning as soil microbes play a fundamental role in biogeochemical cycling and drive productivity in terrestrial ecosystems (van de Heijden et al., 2008). Soil microbial diversity in the Arctic is comparable to that in other biomes (Chu et al., 2010), and the spatiotemporal variability in microbial community composition is large (Lipson, 2007; Blaud et al., 2015; Ferrari et al., 2016). However, it is still uncertain which environmental factors drive the heterogeneity of soil microbial properties in the Arctic.

Altitudinal transects offer a great opportunity to study the distribution of microbial communities adapted to local habitats and explain patterns through natural gradients of soil conditions, presence or absence of vegetation and different climate regimes within short spatial distances (Ma et al., 2004; Körner et al., 2007). Climate change will further affect environmental conditions in the Arctic (Collins et al., 2013), including the expected upward migration of vegetation and increasing plant cover (Vuorinen et al., 2017; Yu et al. 2017). Therefore, the knowledge of current microbial distribution and activity patterns along the altitudinal gradients together with identifying their controlling factors can help to predict the future development of ecosystems in this region. However, such studies are scarce despite the fact that the arctic tundra comprises 5% of the land on Eart′s land surface (Nemergut et al., 2005) and that most coastal areas in the northern circumpolar region have a mountainous character. To date, only few studies assessing altitudinal trends in soil microbial properties have been conducted in the Scandinavian Arctic (Löffler et al., 2008; Männistö et al., 2007). Research on spatial variation in microbial community composition and activity in polar regions was conducted mainly within a narrow elevation range (Oberbauer et al., 2007; Trevors et al., 2007; Björk et al., 2008; Chu et al., 2010; Van Horn et al., 2013; Blaud et al., 2015; Tytgat et al., 2016) or was focused on initial soil development following glacier retreat (Bekku et al., 2004; Yoshitake et al., 2007; Schŭtte et al., 2010). The majority of studies on elevational patterns in microbial community structure (MCS) and activity has been carried out in mountain regions of the lower latitudes from the tropics to the temperate zones. The studies commonly show that microbial activity decreases with increasing elevation (Schinner, 1982 - Alps; Niklińska and Klimek, 2007 - Polish Carpatians; Margesin et al., 2009 - Alps), while there are no general altitudinal patterns in soil microbial diversity and community structure. For example, the microbial community composition did not change along elevational gradients in the Swiss Alps (Lazzaro et al., 2015), while other studies have documented decreasing bacterial (Ma et al., 2004 - western China; Lipson, 2007 - Rocky Mountains; Shen et al., 2013 - northeast China) and fungal (Schinner and Gstraunthaler, 1981 - Alps) diversity with increasing altitude. Several studies reported a mid-altitudinal peak in microbial diversity (Fierer et al., 2011 - Peru; Singh et al., 2012 - Mt. Fuji, Japan; Meng et al., 2013 - central China). Beside the fungal and bacterial diversity, the relative abundance of these main microbial functional groups is also variable. For example, Djukic et al. (2010 - Alps), Xu et al. (2014 - Himalayas) and Hu et al. (2016 - Himalayas) found a decreasing ratio of fungi to bacteria (F/B) with increasing elevation, while Margesin et al. (2009 - Alps) reported the opposite trend in the Central Alps.

Research focusing on environmental controls over microbial communities in polar and alpine regions has recognized many significant factors, including vegetation, litter C/N stoichiometry, organic carbon content, soil pH, nutrient availability, microclimatic conditions, and bedrock chemistry. However, the effect of these variables was site- and scale-specific (Van Horn et al., 2013; Blaud et al., 2015; Ferrari et al., 2016), which highlights the need for further research on environmental controls of microbial community size, activity and structure at local and regional scales. To extend our knowledge about microbial ecology and soil functioning in arctic ecosystems, we conducted a study aiming to assess the activity, biomass and structure of soil microbial communities and to determine their controlling environmental factors along three altitudinal transects located in Central Svalbard. These transects spanned from vegetated tundra habitats in narrow areas at sea level to unvegetated soils at the top of the coastal mountains. The specific objectives of our study were (i) to describe gradients of microclimatic and geochemical soil properties; (ii) to assess microbial activity (soil respiration) and the abundance of main microbial groups (fungi, Gram–negative and Gram–positive bacteria, Actinobacteria, phototrophic microorganisms) using phospholipid fatty acid (PLFA) analysis; and (iii) to identify environmental factors explaining any patterns in soil microbial parameters along these altitudinal gradients.

## 2 Materials and methods

## 2.1    Study area and soil sampling

Petunia Bay (Billefjorden; 78° 40´ N, 16° 35´ E) is located in the center of Svalbard archipelago (Norway) and represents a typical High Arctic ecosystem in the northern circumpolar region. The mean, minimum and maximum air temperatures recorded in the area at 25 m above the sea level (a.s.l.) in the period of 2013 – 2015 were –3.7, –28.3 and 17 °C, respectively. The temperatures stayed permanently below 0 °C for eight months a year (Ambrožová and Láska, 2017). The mean annual precipitation in the Central Svalbard area is only 191 mm (Svalbard Airport, Longyearbyen, 1981–2010) and is equally distributed throughout the year (Førland et al., 2010).

In August of 2012, we collected soils from three altitudinal transects (Tr1 – Tr3) on the east coast of Petunia Bay. Each transect was characterized by four sampling sites at altitudes of 25, 275, 525 and 765 m a.s.l. (± 5 m). The transects were located on slopes with similar exposition (Tr1 W–E, Tr2 WNW–ESE, Tr3 WSW–ENE; Fig. 1) and lithostratigraphy. Soils at the lowest elevations developed from Holocene slope (Tr1 and Tr3) or marine shore deposits (Tr2), while the bedrock at the more elevated sites consisted of dolomite and limestone with units of basal calcareous sandstone (Dallmann et al., 2004). The soils were classified as Leptic Cryosols (Jones et al., 2010) with loamy texture and clay content increasing with altitude (Table 2). Their depth ranged from 0.15–0.2 m to only a few cm at 25 and 765 m a.s.l., respectively. A poorly developed organic horizon was present only at the lowest elevation. The sampling locations were selected in geomorphologically stable areas with similar slopes (20±5°). On each sampling site, nine soil cores (4 cm deep, 5.6 cm in diameter) were collected and then combined, three at a time, into three different mixed samples. Each sample was made up from one soil core taken from the edge of the vegetation tussocks (if vegetation was present) and two cores taken at increasing distance from the vegetation to maintain the consistency with respect to the heterogeneity of vegetation cover and soil surface. The triplicates were collected approximately 5 m apart. Immediately after sampling, the soil was sieved (2 mm) to remove larger rocks and roots, sealed in plastic bags and kept frozen at –20 °C till further processing. Soil subsamples for biomarker analysis were freeze–dried as soon as possible and stored at –80 °C until extraction.

The transects represented climosequences from High Arctic tundra to unvegetated bare soil. Vegetation at the two lowest sites was dominated by *Dryas octopetala*, with significant contribution of *Saxifraga oppositifolia,* and variable contributions of *Cassiope tetragona, Salix polaris* and sedges (*Carex nardina, C. rupestris, C. misandra*; Prach et al., 2012; personal observations). The vascular plants formed scattered patches at 525 m a.s.l. with *Salix polaris* and *Saxifraga oppositifolia* being the most abundant species. The soils at the most elevated sites were covered mainly by soil crusts, with scarce occurrence of *Saxifraga oppositifolia* and *Papaver dahlianum* (personal observations). The percentage cover of the main surface types (i.e. stones, bare soil, vegetation, crusts and mosses) was estimated at each sampling site in an area of from approximately 1m$^2$ in close vicinity to the coring sites (Table S1, Fig. S6).

## 2.2    Monitoring of microclimatic characteristics

To describe the soil microclimatic conditions along the altitudinal transects, we continuously measured soil temperature at –5 cm from 2012 to 2013 directly at the sampling sites of Tr1 using dataloggers (Minikin Ti Slim, EMS Brno, CZ). The soil water content at the time of sampling was determined in soil subsamples by drying to constant weight at 105 °C. The temperature regimes at the respective altitudinal levels were characterized by 10 climatic variables (Table 1). The period of above–zero daily mean ground temperatures is referred to as the summer season throughout the text. We also measured the number of days with daily mean ground temperatures above 5 °C, which characterizes the period suitable for vascular plant growth (Kleidon and Mooney, 2000). The soil surface energy balance was estimated as the sum of daily mean summer temperatures. The records from three years (2011–2013) of continuous measurements at two automated weather stations located at 25 and 455 m a.s.l., approximately 3 km from the transects (hereafter referred as AWS$_{25}$ and AWS$_{455}$, respectively;

Fig. 1; see Ambrožová and Láska, 2017 for a detailed description), were used to evaluate the seasonal variations of soil temperature and moisture regimes (Figs. S2, S3, respectively), and coupling of soil and atmospheric temperatures (measured at -5 cm and 2 m above surface, respectively; Fig. S2). Even though we were not able to continuously measure soil moisture directly at the sampling sites, we regarded data from both AWS locations as representative for the evaluation of seasonal moisture regimes.

## 2.3    Soil characteristics

The particle size distribution was assessed using the aerometric method (Lovelland and Whalley, 2001). Soil type was classified according to the U.S. Department of Agriculture. The soil pH was determined in a soil–water mixture (1:5, w/v) using a glass electrode. The cation exchange capacity (CEC) was considered to be equal to the sum of the soil exchangeable base cations $Mg^{2+}$, $Ca^{2+}$, $Na^+$, $K^+$ extracted with 1M $NH_4Cl$ (Richter et al., 1992). The contribution of $H^+$ and $Al^{3+}$ ions was neglected due to the high soil pH. Base cations accessible for plant and microbial uptake ($Mg^{2+}$, $Ca^{2+}$, $Na^+$, $K^+$) were extracted with Mehlich 3 reagent (Zbíral and Němec, 2000). Cations were measured by atomic absorption spectroscopy (AA240FS, Agilent Technologies, USA). Total soil organic carbon (TOC) and nitrogen (TN) contents were measured in HCl fumigated samples (Harris et al., 2001) using an elemental analyzer (vario MICRO cube, Elementar, Germany).

## 2.4    Microbial respiration

As we were not able to measure soil respiration on site or immediately after soil collection, we measured the potential respiratory activity (soil $CO_2$ production) in a laboratory incubation experiment. We stored and transported the soils in a frozen state because it was previously demonstrated that freezing-thawing has a weaker effect on microbial activity than long-term refrigeration (Stenberg et al., 1998), comparable to that of drying-rewetting (Clein and Schimel, 1994). We then measured microbial respiration in slowly melted field-moist soils twice during the adaptation period (day 4 and 12 of the incubation), which allowed the microbial activity to stabilize after respiratory flushes following sieving pretreatment (Thomson et al., 2010) and freeze-thaw events (Schimel and Clein, 1996), and on day 13, when we expected microbial activity to have settled. Briefly, on day 1, soil subsamples (10 g) were incubated in 100 mL flasks at 6 °C, which corresponds to the mean summer soil temperature of all sites along Tr1. On days 4, 12 and 13, the cumulative $CO_2$ production from the soils was measured using an Agilent 6850 GC system (Agilent Technologies, CA, USA). The flasks were then thoroughly ventilated and sealed again. Due to the high soil pH, the total amount of produced $CO_2$ was corrected for its dissolution and dissociation in soil solution according to the Henderson-Hasselbach equation (Sparling and West, 1990) and expressed as the microbial respiration rate per day.

## 2.5    Microbial biomass and community structure

The soil microbial community structure was determined using PLFA analysis according to a modified protocol of Frostegård et al. (1993). Briefly, 1-3 g (according to TOC content) of freeze–dried soil was extracted twice with a single–phase extraction mixture consisting of chloroform, methanol and citrate buffer. After phase separation overnight, achieved by adding more chloroform and buffer, the organic phase was purified on silica columns (SPE–SI Supelclean 250mg/3 mL; Supelco®, PA, USA) using chloroform, acetone and methanol. The polar fraction was trans–esterified to fatty acid methyl esters (FAME) (Bossio and Scow, 1998). All FAMEs were quantified using methyl-nonadecanoate (19:0) as an internal standard. To identify the FAMEs, retention times and mass spectra were compared with those obtained from standards (Bacterial Acid Methyl Esters standard, 37–component FAME Mix, PUFA–2, and PUFA–3; Supelco, USA). An ISQ mass

spectrometer (MS) equipped with a Focus gas chromatograph (GC) (Thermo Fisher Scientific, USA) was used for chromatographic separation and detection.

Only specific PLFAs were used to assess the microbial community structure: a14:0, i15:0, a15:0, i16:0, i17:0 and a17:0 were used as markers of Gram–positive bacteria (G+); 16:1ω9, 16:1ω5, cy17:0, 18:1ω11, 18:1ω7 and cy19:0 as markers of Gram–negative bacteria (G–); 10Me16:0 and 10Me18:0 as markers of Actinobacteria (Kroppenstedt, 1985), 18:1ω9 and 18:2ω6,9 as fungal markers (Frostegård and Bååth, 1996) and the polyunsaturated fatty acids 18:4ω3and 20:5ω3 were used as markers of phototrophic microorganisms (Hardison et al., 2013; Khotimchenko et al., 2002). The sum of Actinobacterial markers, PLFAs specific to G+ and G– bacteria and general bacterial markers 15:0, 17:0 and 18:1ω5 was used to calculate bacterial biomass and ratio of fungi to bacteria (F/B). The sum of all lipid markers mentioned above and nonspecific PLFAs 14:0, 16:0, 18:0 and 16:1ω7 was used as a proxy for microbial biomass (PLFA$_{tot}$).

## 2.6 Sterol analyses

The β–sitosterol and brassicasterol were used as biomarkers of plant (Sinsabaugh et al., 1997) and microalgal (Volkman, 1986; 2003) residues in organic matter (OM), respectively. Both sterols were determined simultaneously using microwave-assisted extraction adapted from Montgomery et al. (2000) and GC/MS analysis (ISQ MS equipped with Focus GC, Thermo Fisher Scientific, USA). Briefly, 0.5 g of freeze–dried soil was treated with 6 mL of methanol and 2 mL of 2 M NaOH. Vials were heated twice at the center of a microwave oven (2450 MHz and 540 W output) for 25 s. After cooling, the contents were neutralized with 1 M HCl, treated with 3 mL of methanol and extracted with hexane (3×4 mL). Extracts were spiked with an internal standard (cholesterol), evaporated and derivatized by adding of pyridine and 1 % BSTFA at 60 °C for 30 min prior to analysis. The sterols were quantified by internal standard calibration procedure.

## 2.7 Statistical analyses

All data were checked for normality and homoscedasticity, and log–transformed if necessary. The relative PLFA data (mol%) were log-transformed in all statistical tests. The significance of environmental gradients and corresponding shifts in MCS (mol% of summed PLFA specific for fungi, G– and G+ bacteria, Actinobacteria and soil phototrophic microorganisms) in the horizontal (i.e., effect of transect) and vertical direction (i.e., effect of altitude) was tested using partial redundancy analyses (RDA) with covariates. Variation partitioning was subsequently performed to quantify the unique and shared effects of transect and altitude on variability of MCS. A forward selection procedure was used to identify the soil geochemical parameters best explaining shifts in MCS. During the forward selection procedure, only $P$ values adjusted with Holms correction were considered. This procedure is slightly less conservative than the often recommended Bonferroni correction, but it is a sequential procedure and takes into account that the candidate predictors with stronger effects are selected first (Holm, 1979). The multivariate tests were performed without standardization by samples, but with centering and standardization by variables (because the variables were not always measured at the same scale, see Šmilauer and Lepš 2014) and a Monte Carlo test with 1999 permutations. Only the adjusted explained variation is referred to throughout the text. As the triplicate samples cannot be considered as independent observations due to the relatively small distance between samples (otherwise there would be 9 independent transects), only the sampling sites were freely permuted, while the individual samples were exchangeable only within the sampling sites. The differences in particular soil and microbial parameters between the respective transects and altitudes were addressed by ANOVA complemented with Tukey–HSD post hoc tests. Pearson correlation coefficients were used to assess how tightly different variables were related to each other. All statistical tests were considered significant at $P < 0.05$. Multivariate statistical analyses were performed with CANOCO for

Windows version 5.0 (Ter Braak and Šmilauer 2012). For ANOVA, Tukey-HSD tests and correlations between soil and/or microbial parameters, Statistica 13 was used (StatSoft, USA).

## 3 Results

### 3.1 Altitudinal changes in soil microclimate

The soil microclimate at the sites studied was characterized by two distinct periods reflecting air temperature dynamics (compare Fig. S2a with S2b). The winter period typically lasted from the middle of September to early June. The winter soil temperatures were stratified according to elevation and mean temperatures decreased from –4 °C at 25 m a.s.l. to –10 °C at 765 m a.s.l. (Table 1, Fig. S2). By contrast, the short summer period was characterized by a significant diurnal fluctuation of soil temperatures and weak altitudinal temperature stratification (Fig. S2). The length of the summer season more than doubled at the lowest elevations compared with the most elevated study sites. The period with daily mean soil temperatures above 5 °C was shorter by a factor of almost four at the highest elevation. The positive surface energy balance gradually decreased with increasing altitude (Table 1). The maximum daily mean temperatures and diurnal temperature fluctuations were highest at the mid–elevation sites; the highest mean summer soil temperatures were observed at 275 m a.s.l. By contrast, the least and most elevated sites experienced lower summer maximum daily means and soil temperature amplitudes (Table 1). The effect of altitude on soil moisture was significant along Tr1 and Tr3 ($P < 0.001$ and 0.01, F = 22.76 and 7.39, respectively), with soil moisture content decreasing with increasing elevation, but nonsignificant along Tr2. Continual volumetric measurements of soil water content at $AWS_{25}$ and $AWS_{455}$ showed that soil moisture was relatively stable during the summer season, and no desiccation events occured during the summer periods of 2011–2013 (for more information, see Fig. S3).

### 3.2 Gradients of soil geochemical properties and surface vegetation cover

Both factors, transect and altitude, significantly affected soil geochemical properties (partial RDA, pseudo–F = 8.3, $P < 0.001$) and explained 61% of the total variation in soil characteristics. The RDA ascribed most of the explained variability (73%) to vertical zonation. Accordingly, altitude had a significant on all soil parameters (Table 2, 3, Fig. S4), but a significant interactive effect of transect and altitude indicated that the elevational trends were in most cases specific for particular transects (Tables 2, 3). Especially CEC and the availabilities of $Ca^{2+}$, $Mg^{2+}$, $K^+$ and $Na^+$ were spatially variable, reflecting the complicated geology of the Petunia Bay area. The soils along Tr1 were significantly richer in available $Mg^{2+}$ and $K^+$ than soils from the other two transects (Table 2). $Mg^{2+}$ availability also significantly increased with increasing elevation along Tr1 (Table 2). Other soil properties showed more systematic altitudinal patterns. The mean soil pH ranged from 7.8 to 9.0 and increased with altitude along all transects (Table 2, Fig. S4). By contrast, the soil TOC and TN contents declined towards higher elevations along all transects, the exception being the lowest site of Tr2, with lower soil OM content compared with the respective sites from Tr1 and Tr3. The OM-poorest soil occurred at the highest site of Tr1 (Table 3). The soil C/N ratio, sitosterol content in TOC and the ratio between plant–derived sitosterol and brassicasterol of algal origin were solely affected by the altitude. Their values systematically decreased with an increasing elevation irrespective of the soil OM content (Table 3), indicating an altitudinal shift in the OM quality and origin. The percentage of plant cover also continuously decreased with an increasing elevation along Tr1 and Tr3 but was comparable at the three lower sites along Tr2 (Fig. S5), which significantly reflected the trends in soil OM content ($r = 0.53$; $P = 0.001$). Lichenized soil crusts were the

predominant type of soil surface cover at all sites, while mosses covered very small proportions of the surface area. Bare surface without any vegetation (bare soil) occurred only at the two most elevated sites of all transects (Fig. S5, Table S1).

### 3.3 Soil microbial biomass and activity

Soil PLFA content, used here as a measure of soil microbial biomass, was significantly positively correlated with soil TOC and TN contents ($r$ = 0.773 and 0.719, respectively; both $P$ < 0.0001) and soil moisture ($r$ = 0.772; $P$ < 0.0001). It was negatively affected by $Mg^{2+}$ availability ($r$ = -0.775; $P$ < 0.0001). Despite these relations, soil PLFA content did not show any altitudinal pattern. The amounts of soil PLFA were comparable among the differently elevated sites along particular transects (Fig. 2a). Only the most elevated site of Tr1 had significantly lower soil PLFA content than other sites, which corresponded with its very low stock of OM (Table 3). Similarly, neither the flush of microbial respiration measured after soil thawing (day 4 of incubation) nor the respiration measured after stabilization (day 12, not shown, and day13) showed any systematic altitudinal pattern (Fig. 2b, c). Generally, the flush respiration rate was closely related (r = 0.74, P < 0.0001, n = 36) to microbial respiration after stabilization and 2.3 ± 0.3 times higher than the latter, showing a similar freeze-thaw effect on the whole set of samples independently of altitude and transect. The microbial respiration rates measured between days 4 and 12 and after stabilization (day 13), respectively, did not significantly differ in any soil samples. The three lower sites along each transect (from 25 to 525 m a.s.l.) had comparable microbial respiration rates after stabilization. However, the most elevated site of Tr1 had a significantly lower microbial respiration rate, whilst the most elevated sites of Tr2 and Tr3 produced markedly more $CO_2$ than the other sites along these transects (Fig. 2b). The respiration rate was neither related to PLFA nor to TOC contents, but significant positive correlations with soil $Ca^{2+}$ availability and F/B ratio, and a negative correlation with $Mg^{2+}$ availability ($r$ = 0.489, 0.661and -0.545; $P$ = 0.003, < 0.001 and 0.001, respectively) was observed.

### 3.4 Microbial community structure

The factors altitude and transect explained 51% of the total variation in the MCS, with 66 % of the explained variability ascribed to altitude, 26% to transect, and 8% shared by both factors. In fact, the partial RDA a revealed significant interactive effect of both factors on MCS (pseudo–$F$ = 4.8, $P$ < 0.001). The soil geochemical variables explained 72% of the variation in the MCS (pseudo–$F$ = 7.1; $P$ < 0.001), indicating that the separate and interactive effect of altitude and transect on MCS was largely driven by vertical and horizontal variability in soil properties. The forward selection of explanatory variables retained four geochemical parameters: $Mg^{2+}$ availability, pH, moisture and TOC content, all together accounting for 55% of variation in the data (pseudo–$F$ = 11.6, $P$ < 0.001). The most pronounced shift in the MCS was caused by different altitudinal preferences of bacteria and fungi. Bacteria were consistently more abundant in the soils from lower elevations, having lower pH and higher TOC and moisture contents (Fig. 3). In general, PLFAs specific to G– bacteria were more abundant than PLFAs of G+ bacteria (Fig. 4a; mean G–/G+ ratio ± SD = 1.76 ± 0.17; n = 36). By contrast, the fungal contribution to the microbial community increased with increasing altitude, at the sites with TOC poorer soils and higher pH (Fig. 3). Therefore, the F/B ratio gradually increased with increasing altitude along all three transects (Fig. 4b). The significant interactive effect of altitude and transect on MCS was mainly connected to a strong effect of soil $Mg^{2+}$ availability, which was higher along the whole Tr1 and differentiated its microbial communities from the sites of Tr2 and Tr3, where microbial communities of corresponding sites were more similar. The differences in MCS among the respective sites along Tr1 on the one hand and the other two transects on the other further increased towards higher elevations, together with increasing soil $Mg^{2+}$ availability along Tr1 (Fig. 3). As a result, the soil with the poorest TOC content and highest $Mg^{2+}$ concentration at the highest site on Tr1 had the most distinct MCS of all sites. Its microbial community was characterized by higher abundance of Actinobacteria and PLFAs of phototrophic microorganisms and a much lower contribution of G– bacteria compared with the communities at all other sampling sites (Fig. 3, 4a).

## 4        Discussion

Our measurements showed that soils along an elevation gradient from 25 to 765 m a.s.l. face different microclimatic regimes. During the winter period, soil temperatures were relatively stable but significantly stratified according to altitude (Fig. S1, S2a). The significant decrease of mean winter soil temperatures with increasing elevation (Table 1) can strongly reduce winter soil microbial activity at high altitudes compared with the least elevated sites (Drotz et al., 2010; Nikrad et al., 2016). The comparison of summer temperatures and their fluctuations indicated that both the lowest and highest sites experienced on average a colder, but more stable soil microclimate during summer compared with the mid-elevation sites. The significantly longer summer season, increasing number of days with mean temperature above 5 °C and a rising positive surface energy balance with decreasing elevation (Table 1) positively affected the occurrence and proliferation of vascular plants (Kleidon and Mooney, 2000; Klimeš and Doležal, 2010), which had strong implications for the change in edaphic conditions along transects. The pronouncedly greater plant cover at lower elevations (Table S1) resulted in increased litter inputs and greater stocks of soil OM with higher C/N ratio (Table 3). Plant growth was also associated with decreasing soil pH via root respiration, cation uptake and release of $H^+$ and organic acids from roots (van Breemen at al., 1984). The increasing soil OM content was further positively related to soil moisture (Fig. 3).

The altitudinal shifts in soil edaphic properties were not significantly reflected in soil microbial biomass and potential microbial respiration. Soil PLFA contents were generally comparable between all sites along any particular elevation transect, with the exception of the very low soil PLFA concentration at the highest site of the Tr1 (Fig. 2a). There are no other studies from High Arctic ecosystems reporting an altitude effect on soil microbial biomass. However, other studies conducted on alpine gradients in the temperate and boreal zones documented weak or absent altitudinal trends in the microbial biomass (Djukic et al., 2010, and Xu et al., 2014, using PLFA; Löffler et al., 2008, using cell counts) but also a negative effect of elevation in the Alps (Margesin et al., 2009) and northern Finland (Väre et al., 1997). Importantly, none of the studies considered unvegetated habitats and all of them were conducted in soils with acidic or neutral soil pH.

Microbial respiration did not change systematically with increasing elevation. The three lowest sites along each transect always had comparable soil microbial respiration rates (Fig. 2b), while soil microbial activities at the highest sites differed. The most elevated site on Tr1 showed significantly lower respiration rates than the lower sites on this transect, which was in line with the lowest OM content and soil PLFA content. However, the soils from the highest sites on both Tr2 and Tr3 respired significantly more than the soils from lower sites on these transects, irrespective of the relatively stable microbial biomass along these transects. This is in contrast to other studies, which reported decreasing microbial activity with increasing elevation (Schinner, 1982 - Alps; Väre et al., 1997 - northern Finland; Niklińska and Klimek, 2007 - Polish Carpatians). However, these studies were conducted at lower latitudes and the altitudinal gradients studied did not include unvegetated habitats.

Microbial activities in this study were measured in sieved freeze-stored and not in intact fresh samples (see section 2.3 for details). The respiration rates measured after thawing were necessarily affected by sample handling and show the potential activity of soil microbial communities in the soils. The *in situ* microbial respiration could differ from those values as sieving and freezing-thawing has been shown to affect soil C and N fluxes (Hassink et al., 1992) and increase respiration rates (Thomson et al., 2010). The effect of sieving on measured soil respiration could be stronger in samples from the most elevated sites due to disruption of the vertical organization of the biological soil crusts (Belnap and Lange, 2003) compared with the more homogeneous soils from lower altitudes. However, the respiration rates in three subsequent measurements (after flush – day 4, after adaptation – day 12, and after stabilization – day 13) were positively correlated (r = 0.93 and 0.74,

both P < 0.0001, n = 36), the ratios between the flush and stabilized respiration rates were comparable across all soils (compare Fig. 2b and 2c), and the differences in microbial activities between sites described above were consistent. Our data are also in agreement with the study of Larsen et al. (2002), who found a comparable response of microbial activity to freeze-thaw events in two different arctic ecosystem types. We thus suggest that the soils responded similarly to the storage treatments, independently of site location, and that the observed differences in potential soil microbial activities are representative for the transects studied. Therefore, the higher soil microbial respiration at the most elevated sites points to a greater lability of the present OM (Lipson et al., 2000 - Rocky Mountains; Uhlířová et al., 2007 - Siberian tundra) and/or to a shift in microbial communities towards groups with higher potential to mineralize the OM (Gavazov, 2010 - Alps; Djukic et al., 2013 - Alps). Previous studies, considering either bare soil or vegetated habitats, reported an increasing complexity of soil OM with elevation (Ley et al., 2004 - Rocky Mountains; Xu et al., 2014 - Himalayas). However, in this study the bulk of OM and microbial biomass at the most elevated sites was associated with biological soil crusts with high algal and cyanobacterial abundance (Table S1, Fig. S5), known for their high microbial activity (Pushkareva et al., 2017 - Svalbard; Bastida et al., 2014 - Spain). The high microbial activity in the most elevated sites could be ascribed to a prevalence of compounds of algal/cyanobacterial origin with very low proportion of complex and slowly decomposable compounds and protective waxes (like cutin and suberin) mainly derived from vascular plants. In accordance with this, the sitosterol to brassicasterol ratio gradually decreasing with increasing elevation (Table 3) and the increasing sitosterol content in the TOC pool at lower elevations point to a growing importance of microalgal sources of OM in high elevation habitats (Sinsabaugh et al., 1997; Rontani et al., 2012). Even though both sterols can be found in higher plants and microalgae, the changing ratio indicates a shift in the origin of OM (reviewed by Volkman, 1986, see also Volkman, 2003). Changes within microbial communities, which can also help to explain higher soil microbial respiration at the most elevated sites, are discussed below.

While the soil PLFA content did not change along the elevation transects studied, we found a systematic altitudinal shift in PLFA composition, resulting in a significantly increasing F/B ratio towards higher elevations. This shift was best explained by a decreasing soil OM content and soil moisture and increasing pH (Fig. 3). Reports about soil F/B ratios and their altitudinal changes from the High Arctic are missing, but studies from lower latitudes showed either a similar trend of increasing F/B ratio with an altitude in the Alps (Margesin et al., 2009) or the opposite altitudinal effect in the Alps (Djukic et al., 2010) and Himalayas (Xu et al., 2014; Hu et al., 2016). Such divergent results indicate that altitude alone is not the key driving factor of the soil F/B ratio. In contrast to our observations, these studies reported very low soil F/B ratios of 0.05-0.2, which may indicate an important role of fungi in the functioning of arctic habitats. Soil pH has previously been identified as the main driver of fungal vs. bacterial dominance in the soil (Baath and Anderson, 2003; Högberg et al., 2007; Rousk et al., 2009; Siles and Margesin, 2016). Fungi have been found to be more acid-tolerant than bacteria, leading to higher F/B ratio in acidic soils (Högberg et al., 2007; Rousk et al., 2009; reviewed by Strickland and Rousk, 2010). However, here we report high F/B ratios in alkaline soils (pH 7.8-9.0) and an increase in F/B ratios with increasing soil pH. A similar trend was reported by Hu et al. (2016), but the authors found F/B ratios one order of magnitude lower than in our study. A possible explanation of the generally high fungal abundance and increasing F/B ratio with elevation could be a greater competitiveness of fungi compared to bacteria at sites with highly alkaline soil pH and severe winter microclimate due to their wider pH (Wheeler et al., 1991) and lower temperature (Margesin et al., 2003) growth optima. We further found that the increasing F/B ratio was significantly coupled with an increasing soil respiration (r = 0.649; P < 0.001). Indeed, such a relationship may be related to a greater fungal ability either to grow in the soil conditions at the most elevated sites or to utilize available C sources more efficiently (Ley et al., 2004; Bardgett et al., 2005; Nemergut et al., 2005; van der Heijden et al., 2008). In turn, the higher bacterial contribution at lower elevations may be associated with a bacterial preference for utilization of labile root exudates released by vascular plants (Lipson et al., 1999; Lipson et al., 2002) and more benign soil conditions represented by higher moisture contents and less alkaline soil pH. However, direct linking of potential activities measured in the incubation experiment with in situ abundance of bacteria and fungi may be problematic as the handling of

sample could alter the original MCS (Petersen and Klug, 1994). As the projected warming in the Arctic (Collins et al., 2013) will likely cause an upward migration of the vegetation and increasing plant cover to the detriment of lichens and biological soil crusts (Vuorinen et al., 2017; Yu et al. 2017; de Mesquita et al., 2017), the soil microbial communities could respond by decreasing their F/B ratios at higher elevations.

Apart from the systematic altitudinal shift in the F/B ratio connected mostly with the plant occurrence and its effect on soil edaphic conditions, we observed a strong shift in the bacterial composition, which differentiated the altitudinal trends in the soil MCS along Tr1 from trends along Tr2 and Tr3. This difference between transects increased towards higher elevations and was best explained by $Mg^{2+}$ availability. The character of the parent substrate thus mostly controlled soil microbial properties at the most elevated sites, which generally had low OM content (Fig. 3). The soils from Tr1, except at the lowest site, had lower G– to G+ bacterial ratios within microbial communities than soils from the other two transects. Further, the microbial community of the most elevated site of Tr1 showed greater abundance of actinobacteria and phototrophic microorganisms than all other sites (Fig. 3, 4a). This site was the most extreme habitat among all the sites studied, with the highest proportion of bare, unvegetated soil surface (Fig. S5), the lowest OM and moisture contents, the highest soil pH and $Mg^{2+}$ availability and consequently also the most distinct microbial characteristics (Fig. 2, 3). It is known that high $Mg^{2+}$ availability inhibits growth of many soil bacterial species. The inhibitive $Mg^{2+}$ levels observed were 5 and 50 ppm for G– and G+ bacteria, respectively (Webb 1949), indicating that these bacterial groups significantly differ in their tolerance for enhanced $Mg^{2+}$ levels. Assuming that half of the available $Mg^{2+}$ was in soil solution and the average soil moisture content was 20%, the $Mg^{2+}$ concentrations would have ranged approximately from 16 to 140 ppm, which could explain the decreased abundance of G- bacteria in sites with high $Mg^{2+}$ availability. This inhibitive $Mg^{2+}$ effect is also in accordance with the negative correlations between $Mg^{2+}$ availability and soil microbial biomass and respiration found in our study, and could explain the lower microbial biomass and respiration in the soils from Tr1. Our data thus indicate that, besides the traditionally identified drivers of microbial activity and MCS such as soil OM content, moisture and pH, $Mg^{2+}$ availability is an important factor in shaping the microbial environment along arctic altitudinal transects on dolomitic parent materials.

## 5    Conclusions

The results obtained in this study have shown significant altitudinal zonation of most edaphic properties, but also significant horizontal heterogeneity, resulting in transect-specific effects of altitude on abiotic soil properties. Our data demonstrated that soils on the most elevated, unvegetated sites around Petunia Bay can host microbial assemblages comparable in size and activity with those of the tundra. The high microbial biomass and activity at the most elevated sites were almost exclusively associated with biological soil crusts, largely contributed by fungi. However, their development was impeded on some sites by high pH, low moisture and high $Mg^{2+}$ availability, resulting in pronouncedly low OM content, microbial biomass and a distinct MCS. Despite the ubiquitous occurrence of soil crusts, the gradual increase in plant productivity and litter input with decreasing elevation was associated with decreasing soil pH, increasing OM content and soil moisture. The soil edaphic and microbial properties became less spatially variable with increasing OM content. As the rise in temperatures and humidity predicted by climatic models will likely cause an upward migration of the vegetation and increasing plant cover, the greater inputs of plant litter will overcome the dominant influence of parent material and lead to an increasing abundance of bacteria and a decreasing F/B ratio in the summer microbial assemblages.

## Author´s contributions

P. Kotas and E. Kaštovská analyzed the data and wrote the manuscript with the assistance of all coauthors. P. Kotas and J. Elster designed the study and carried out sampling. The microbial community structure and environmental parameters were assessed by P. Kotas, E. Kaštovská and H. Šantrůčková.

## Acknowledgements

The field work on Svalbard was funded by the Ministry of Education, Youth and Sports of the Czech Republic through CzechPolar – Czech polar stations, construction and logistic expenses (LM2010009 and RVO67985939). The lab work was supported by the SoWa Research Infrastructure funded by the Ministry of Education, Youth and Sports of the Czech Republic, program "Projects of Large Infrastructure for Research, Development, and Innovations" (grant LM2015075). We also thankJan Kavan for help with logistics during the field work. The authors thank Gerhard Kerstiens for language editing and reviewers for their valuable comments.

## Competing interests

The authors declare that they have no conflict of interest.

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

**Tables**

**Table 1. Climatic variables; temperatures given in °C**

| Site Altitude [m a.s.l.] | Means Summer | Means Winter | Means Year | Min daily means Winter | Max daily means Summer | Mean daily amplitude Summer | Max daily amplitude Summer | Number of days with daily mean > 0 °C | Number of days with daily mean > 5 °C | Positive soil surface energy balance |
|---|---|---|---|---|---|---|---|---|---|---|
| 25 | 5.8 | -3.6 | -0.8 | -7.0 | 11.2 | 5.2 | 10.9 | 110 | 62 | 615 |
| 280 | 7.1 | -5.7 | -2.7 | -10.3 | 14.5 | 8.5 | 18.2 | 96 | 54 | 571 |
| 520 | 5.8 | -8.9 | -4.9 | -15.8 | 14.7 | 8.1 | 17.7 | 91 | 40 | 480 |
| 765 | 5.3 | -9.5 | -6.6 | -17.1 | 11.6 | 5.5 | 14.0 | 51 | 11 | 290 |

























Table 2. Geochemical characteristics of soils along three altitudinal transects (Tr1 - Tr3). Means ± SD (n = 3) are given in the upper part of the table. Results of two–way ANOVA (F–values) of the effects of transect (Tr), altitude (Alt) and their interaction (Tr x Alt) are presented in the lower part of the table.

| transect | altitude [m a.s.l.] | soil type | soil moisture [%] | pH | CEC [meq/100g$^{-1}$] | $Ca^{2+}$ [mg g$^{-1}$] | $Mg^{2+}$ [mg g$^{-1}$] | $K^+$ [µg g$^{-1}$] | $Na^+$ [µg g$^{-1}$] |
|---|---|---|---|---|---|---|---|---|---|
| Tr1 | 25 | sandy loam | [a] 28.4 ± 2.5 | [b] 7.8 ± 0.1 | [a] 35.8 ± 0.4 | [b] 4.9 ± 0.2 | [c] 0.50 ± 0.03 | [b] 104 ± 2.3 | [a] 16.0 ± 1.4 |
| | 275 | sandy loam-loam | [b] 18.0 ± 0.5 | [b] 7.9 ± 0.2 | [b] 27.4 ± 2.3 | [b] 5.2 ± 0.6 | [c] 0.55 ± 0.08 | [b] 81 ± 8.8 | [bc] 8.4 ± 1.3 |
| | 525 | loam | [b] 18.6 ± 2.5 | [b] 8.1 ± 0.1 | [b] 30.3 ± 0.7 | [b] 4.3 ± 0.4 | [b] 0.85 ± 0.04 | [a] 160 ± 18.1 | [b] 11.3 ± 1.1 |
| | 765 | clay-loam | [c] 12.1 ± 1.8 | [a] 9 ± 0.0 | [b] 26.8 ± 2.3 | [a] 19.8 ± 1.0 | [a] 1.25 ± 0.06 | [c] 11 ± 2.7 | [c] 7.3 ± 0.0 |
| Tr2 | 25 | sandy loam | [a] 21.1 ± 2.4 | [c] 7.8 ± 0.1 | [b] 25.6 ± 2.7 | [b] 14.7 ± 2.6 | [c] 0.19 ± 0.01 | [ab] 52 ± 4.0 | [a] 13.2 ± 1.7 |
| | 275 | sandy loam-loam | [a] 21.1 ± 2.4 | [c] 7.9 ± 0.1 | [b] 30.3 ± 1.7 | [ab] 16.5 ± 1.1 | [b] 0.26 ± 0.01 | [a] 59 ± 4.3 | [ab] 10.1 ± 1.7 |
| | 525 | sandy loam-loam | [a] 21.7 ± 5.3 | [b] 8.4 ± 0.1 | [b] 30.8 ± 1.1 | [c] 7.8 ± 1.6 | [a] 0.34 ± 0.01 | [a] 69 ± 3.3 | [ab] 9.6 ± 1.8 |
| | 765 | loam | [a] 22.5 ± 1.7 | [a] 8.8 ± 0.1 | [a] 45.1 ± 0.5 | [a] 27.9 ± 9.3 | [b] 0.25 ± 0.01 | [b] 41 ± 8.8 | [b] 8.1 ± 1.4 |
| Tr3 | 25 | sandy loam | [a] 39.5 ± 1.4 | [b] 8.1 ± 0.1 | [a] 49.4 ± 2.1 | [c] 7.7 ± 0.3 | [a] 0.20 ± 0.03 | [b] 52 ± 5.3 | [a] 17.1 ± 1.1 |
| | 275 | sandy loam-loam | [ab] 31.9 ± 2.9 | [b] 8.1 ± 0.1 | [b] 39.2 ± 5.4 | [b] 10.8 ± 0.6 | [a] 0.21 ± 0.01 | [ab] 59 ± 1.9 | [a] 18.5 ± 0.5 |
| | 525 | loam | [ab] 28.2 ± 6.5 | [b] 8 ± 0.1 | [b] 34.9 ± 3.0 | [ab] 13.0 ± 4.6 | [a] 0.22 ± 0.00 | [a] 66 ± 6.6 | [a] 18.4 ± 3.1 |
| | 765 | loam | [b] 22.5 ± 1.7 | [a] 8.8 ± 0.1 | [b] 30.6 ± 3.9 | [a] 14.2 ± 0.1 | [b] 0.16 ± 0.00 | [b] 52 ± 1.6 | [b] 9.9 ± 0.2 |
| | d.f. | | | | | | | | |
| Tr | 2 | | **31.4 \*\*\*** | 0.10 | **22.1 \*\*\*** | **6.43 \*\*** | **634 \*\*\*** | **51.7 \*\*\*** | **36.2 \*\*\*** |
| Alt | 3 | | **11.1 \*\*\*** | **98 \*\*\*** | **4.61 \*** | **14.1 \*\*\*** | **66.9 \*\*\*** | **74.9 \*\*\*** | **18.7 \*\*\*** |
| Tr x Alt | 6 | | **5.07 \*\*** | **5.6 \*\*\*** | **20.5 \*\*\*** | 0.83 | **60.6 \*\*\*** | **31.6 \*\*\*** | **3.94 \*\*** |

Different letters indicate significant differences between sampling sites along particular transects ($P < 0.05$; upper part of the table). Statistically significant differences are indicated by: * $P < 0.05$, ** $P < 0.01$, *** $P < 0.001$ (lower part of the table).

**Table 3.** Total soil carbon (TOC) and nitrogen (TN) contents, their molar ratios, contents of sitosterol in TOC and sitosterol / brassicasterol ==ratios in== soils along three altitudinal transects (Tr1 - Tr3). Means ± SD (n = 3) are given in the upper part of the table. Results of two–way ANOVAs (F–values) of the effects of transect (Tr), altitude (Alt) and their interaction (Tr x Alt) are presented in the lower part of the table.

| transect | altitude | TOC | TN | TOC/TN | Sitosterol | Sitosterol / Brassicasterol |
|---|---|---|---|---|---|---|
| | [m a.s.l.] | [mg g$^{-1}$] | [mg g$^{-1}$] | | [µg g$^{-1}$ TOC] | |
| Tr1 | 25 | $^c$ 70.6 ± 13.4 | $^b$ 5.0 ± 1.01 | $^b$ 12.1 ± 0.2 | $^c$ 534 ± 62.8 | $^b$ 5.5 ± 0.4 |
| | 275 | $^b$ 21.1 ± 1.9 | $^a$ 2.0 ± 0.29 | $^{ab}$ 9.0 ± 0.7 | $^{bc}$ 521 ± 140 | $^b$ 5.3 ± 0.8 |
| | 525 | $^b$ 18.5 ± 4.2 | $^a$ 1.8 ± 0.31 | $^{ab}$ 8.8 ± 0.7 | $^{ab}$ 293 ± 66.5 | $^b$ 4.7 ± 1.0 |
| | 765 | $^a$ 4.4 ± 1.5 | $^a$ 0.5 ± 0.07 | $^a$ 7.9 ± 2.6 | $^a$ 81.1 ± 2.7 | $^a$ 2.3 ± 0.4 |
| Tr2 | 25 | $^{ab}$ 30.6 ± 4.8 | $^a$ 1.9 ± 0.40 | $^c$ 13.7 ± 0.9 | $^{bc}$ 515 ± 44.9 | $^b$ 6.7 ± 0.7 |
| | 275 | $^b$ 37.2 ± 5.0 | $^a$ 3.0 ± 0.26 | $^b$ 10.7 ± 0.7 | $^c$ 616 ± 143 | $^b$ 5.6 ± 1.2 |
| | 525 | $^a$ 24.4 ± 7.8 | $^a$ 1.9 ± 0.64 | $^b$ 9.8 ± 1.2 | $^{ab}$ 299 ± 73.3 | $^a$ 2.9 ± 0.4 |
| | 765 | $^a$ 21.6 ± 3.6 | $^a$ 2.8 ± 0.20 | $^a$ 6.7 ± 0.6 | $^a$ 161 ± 36.9 | $^a$ 2.7 ± 0.7 |
| Tr3 | 25 | $^c$ 81.1 ± 8.7 | $^b$ 6.1 ± 0.38 | $^b$ 11.5 ± 0.7 | $^b$ 587 ± 144 | $^b$ 6.4 ± 2.1 |
| | 275 | $^b$ 62.2 ± 9.1 | $^{ab}$ 4.8 ± 0.32 | $^b$ 11 ± 0.7 | $^{ab}$ 370 ± 42.9 | $^a$ 4.2 ± 0.7 |
| | 525 | $^{ab}$ 39.6 ± 11.4 | $^a$ 4.8 ± 0.32 | $^b$ 10.6 ± 0.6 | $^a$ 270 ± 112 | $^a$ 3.3 ± 1.0 |
| | 765 | $^a$ 23.1 ± 3.9 | $^a$ 2.5 ± 0.37 | $^a$ 7.9 ± 0.2 | $^a$ 151 ± 37.8 | $^a$ 3.1 ± 0.9 |
| | d.f. | | | | | |
| Tr | 2 | **27.8 \*\*\*** | **31.5 \*\*\*** | 1.57 | 0.79 | 1.04 |
| Alt | 3 | **42.4 \*\*\*** | **26.4 \*\*\*** | **23.6 \*\*\*** | **28.4 \*\*\*** | **14.4 \*\*\*** |
| Tr x Alt | 6 | **8.33 \*\*\*** | **11.3 \*\*\*** | 1.96 | 1.34 | 2.17 |

Different letters indicate significant differences between sampling sites along particular transects ($P < 0.05$; upper part of the table). Statistically significant differences are indicated by: * $P < 0.05$, ** $P < 0.01$, *** $P < 0.001$ (lower part of the table).

**Figures**


Figure 1. Location of the three transects investigated (Tr1 - Tr3) and automated weather stations (AWS) in Petunia Bay (Petuniabukta), Billefjorden, Central Svalbard. Map source: map sheet C7, Svalbard 1:100 000, Norwegian Polar Institute 2008.



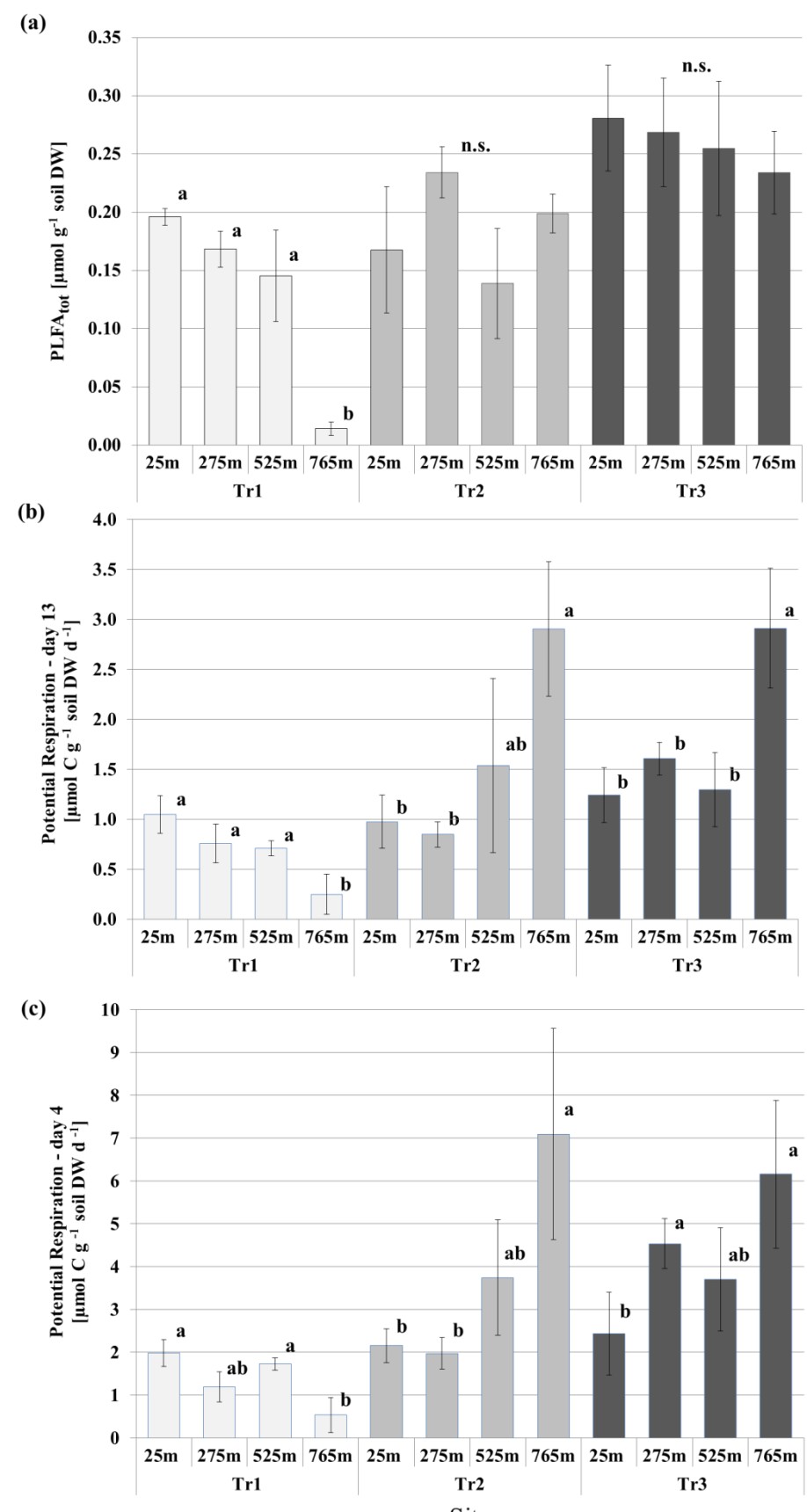


Figure 2. The soil PLFA contents (a), potential respiration rates measured at day 13 (b) and at day 4 (c) in soils from different elevations along three altitudinal transects (Tr1 - Tr3). Error bars indicate mean ± SD (n = 3). Lower case letters denote significant differences between altitudes within particular transects ($P < 0.05$; one–way ANOVA combined with Tukey post-hoc test).

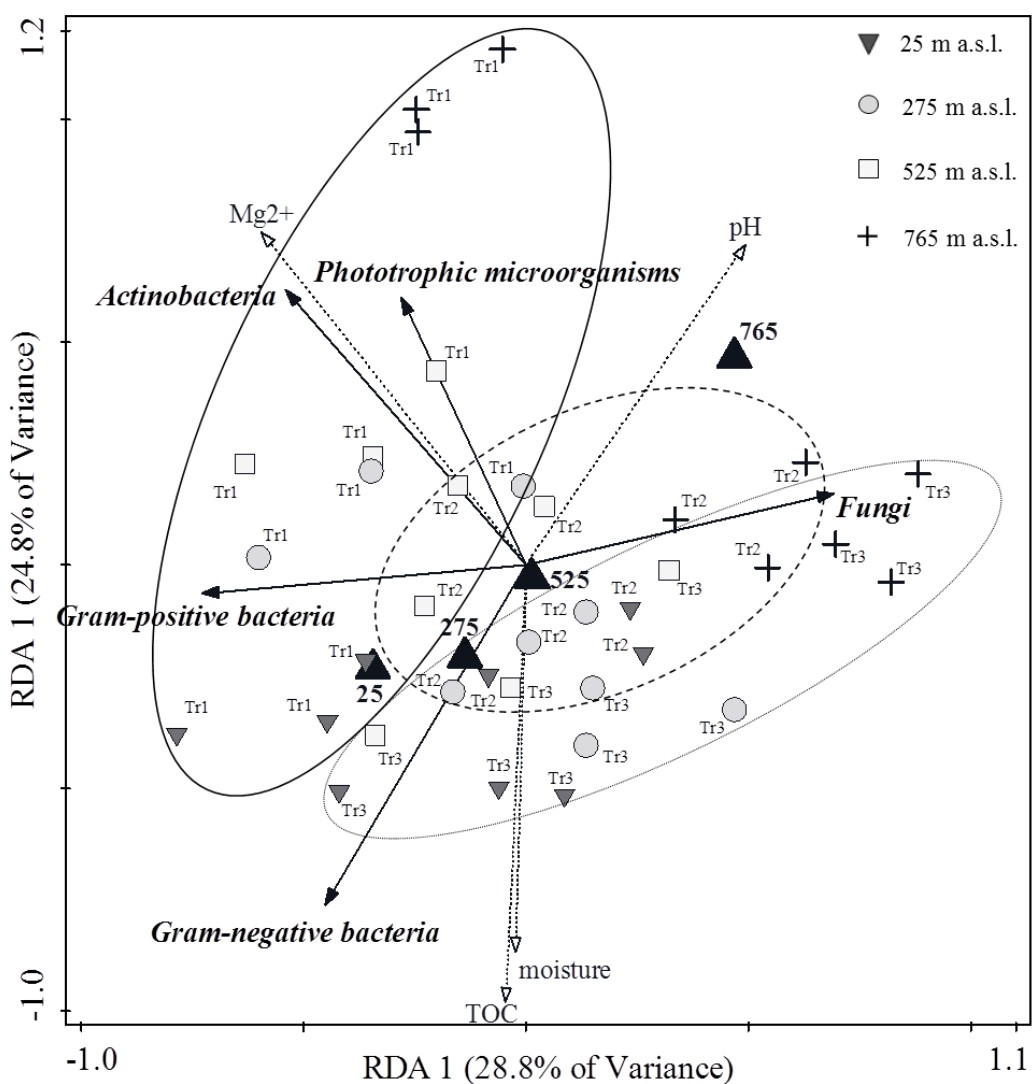

Figure 3. Correlation between the abundance of the main microbial groups (bold italic) and the soil geochemical parameters that were retained by forward selection from all explanatory variables collected. The altitude of sampling sites was used as a supplementary variable. *Arrows* indicate the direction in which the respective parameter value increases, *solid* lines indicate microbial groups, *dotted* lines indicate environmental variables retained by the forward selection procedure. *Upright triangles* are centroids of sites with corresponding elevation (n = 9), numbers indicate elevation (m a.s.l.). The thin solid line encases sites along Transect 1 (Tr1), the dashed line encases sites along Transect 2 (Tr2), and the dotted line encases sites along Transect 3 (Tr3). The numbers in parentheses are the portions of the variation explained by each axis.

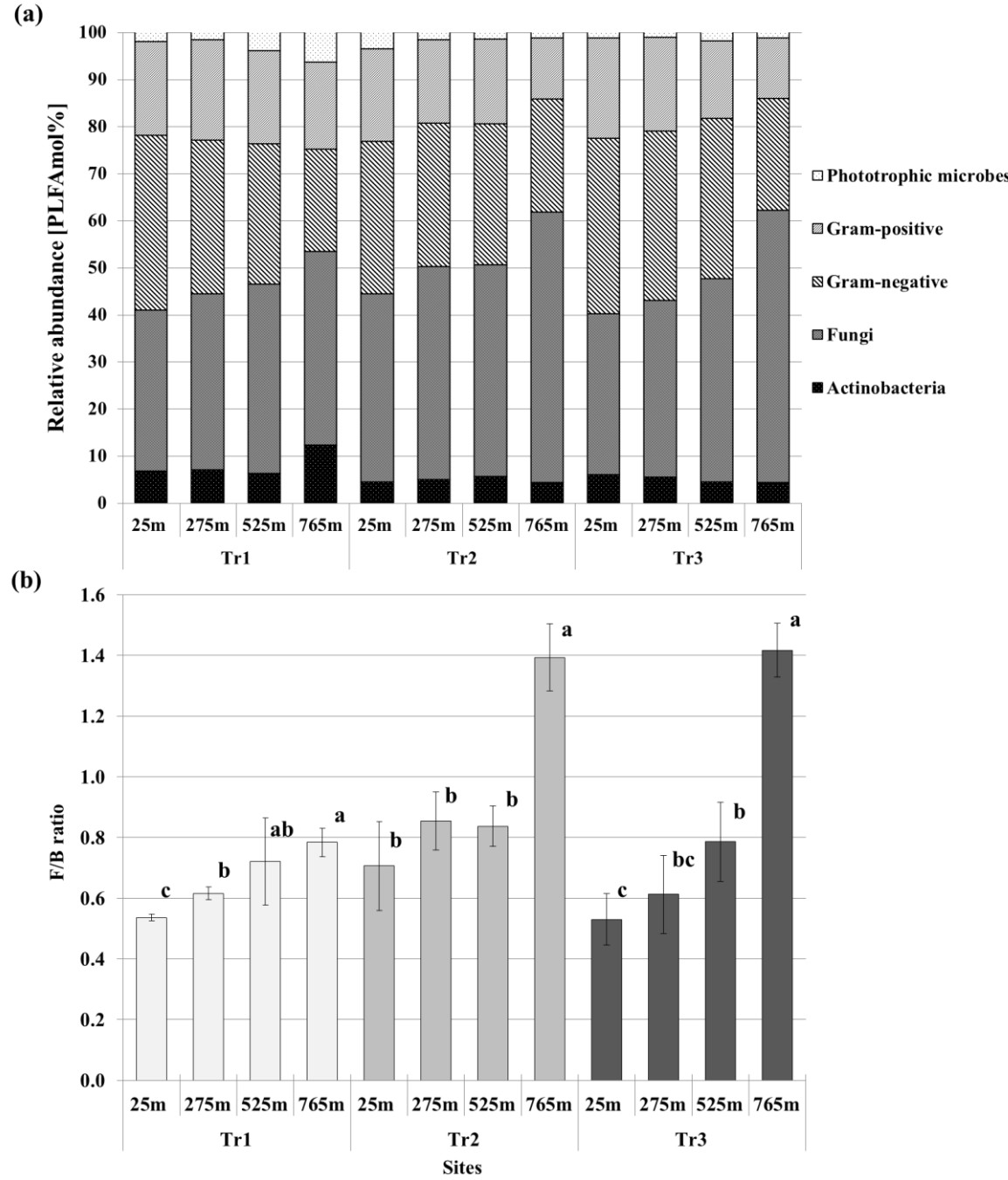

**Figure 4. Relative abundance of specific PLFAs within the microbial community (a) and fungi to bacteria (F/B) ratios (b) along**
**altitudinal transects (Tr1-Tr3). Error bars indicate mean ± SD (n = 3). Lower case letters denote significant differences between**
**altitudes within particular transects ($P < 0.05$; one–way ANOVA combined with Tukey post hoc test).**
