# Peer review of "Soil microbial biomass, activity and community composition along altitudinal gradients in the High Arctic (Billefjorden, Svalbard)"

_Biogeosciences, 2017_

## Referee Comment (RC1) · Anonymous Referee #1 · 10 Aug 2017

General comments The authors investigate the effect of horizontal (across a valley) and vertical (altitude) gradients on microbial community structure (PLFA), biomass and activity in High Arctic. They found that both gradient affect microbial parameters, with shift in the dominance of bacteria and fungi related to the chemistry of the bedrock. The study target interesting question, is relevant for publication in Biogeosciences and is overall well done. My main criticisms are the method used to measure microbial activity (main issue), and too many assumptions made outside the variables measured, going beyond what the results can show.

The measure of microbial activity seems unrealistic. First, 2 mm soil was used which

was frozen and thaw prior to incubation in the lab. So, the microbial community and soil endure 1 freeze-thaw cycle + sieving that will affect OM availability and the microbial community. Then you left the samples for 14 days at 6 degrees before measuring the $CO_2$ for 24h, which you define as "basal respiration". Fourteen days represent $\frac{1}{4}$ of the summer (> 5 degrees) in the Arctic (low altitude) or even your entire summer for the high-altitude site (Table 1), this is a significant amount of time in the Artic. There is no justification and references used to explain why you made these choices. Overall, we can doubt that the high altitude produce high $CO_2$ emissions in in-situ conditions and we can ask the values of your results regarding microbial activity. You did not discuss at any time the limitations of such measurement. We can imagine that the microbial community adapted better or took longer to adapt to incubation condition in high altitude soil explaining the higher $CO_2$ emissions at 14 days. We can also imagine that at low altitude, because of the higher TOC, the $CO_2$ emissions are high rapidly after thawing and after 14 there is not much activity, while for high altitude it took longer to mineralise more complex OM. In other words, your results of microbial activity could be just the results of your incubation/sample preparation. You need to fully acknowledge this in the article, and avoid any conclusion stating that high altitude is a hot spot of microbial activity because your data can't fully support this. You need to be much more conscious about the microbial activity result. Have you measure $CO_2$ emissions over time?

The discussion and conclusions are too long and go far beyond what you can say based on your results. There are many sections you discuss about the dynamic of microbial community but you only did one sampling time. You can't make big conclusions about dynamic of the system, such as L362-376. You need to just briefly mention potential dynamic but don't go much further. Similarly, you speak a lot about the effect of plant cover even you did not measure any parameters to characterize the plant cover (you also forgot to mention anything about mosses and lichens despite their importance in the Arctic) such as above ground biomass, root biomass, percentage cover (did you properly assessed it?), diversity. You just described the main vascular plants. So your section 4.4, is simply too long and not fully supported with your data. This entire section

could be reduced in few sentences and focus on presence/absence of plants and not linked to microbial dynamic.

On the other hand, your discussion lack of putting your results in perspective with the literature, for example other studies investigating microbial community in bare/unvegetated soil (there is several article on this in the Arctic). It would be interesting to see if F/B ratio is similar in unvegetated a low altitude compared to other study. Your main conclusion should be the absence of plant rather than altitude effect, especially if we consider that high altitude soil in the Arctic (especially High Arctic) is likely to be rare, and also extremely shallow (only few cm) and not a massive stock of C. So we can wonder of their importance? Also you should more conclude on the site effect you have (Gr1) as site is clearly an effect on microbial biomass (consistently lower) and structure for the high altitude site.

Another point you miss in your discussion is the fact that the soil you study is always alkaline (are alkaline soils prevalent on Svalbard or acidic?). You need to discuss or conclude if your results would be similar on acidic soil in the Arctic and compare with the relevant literature as pH is a big driver of microbial community including in the Arctic. This is an important point to make and be more critical about your results.

Finally, you find an effect of altitude on microbial biomass mainly when you divide the data by TOC. This bias the data and don't reveal hidden effect of altitude. Yes, you have less C at high altitude but you don't have more microbial biomass. The fact there is more biomass per unit of C is not of a major interest. Dividing your results by TOC is not important and bias your results. Focus on the altitude effect on microbial community structure and ratio, and acknowledge that there is not a major effect of altitude on biomass but rather a site effect.

Specific comments Altitude and transect are both gradient and not only transect. This is really confusing in the text when you speak about gradient, as it is unclear if you speak about vertical or horizontal. You can't refer to the horizontal gradient as "gradient" and

the vertical one as altitude. Decide if you speak about vertical or horizontal gradient, or altitude and transect. Change in the entire text, but don't go from "gradient" for horizontal and then use gradient also for "vertical". Be consistent.

Introduction L20: true but it is simply related to less C from a plant origin, it does not mean that there is more biomass. This is not true as well for GR1. You are telling only one part of the story here. L21: "the 2 dominant microbial groups" it sounds like it is unusual or a result on its own but in the same time with PLFA you have access to fungi and bacteria only. Just speak about fungi and bacteria. L23: you didn't measure microbial dynamic over time, only the change in soil temperature. So, I would focus on what you measured and not make assumption, especially in the abstract. Keep this for the discussion L25-26: the conclusion is an overstatement. In general, unvegetated area should be considered as previous studies showed. Speaking about high elevation as hotspots of microbial activity based on 1 measurement is an overstatement (see main comment). L36-41: this is normal as high altitude usually have no soil present or are extremely shallow (few cm) and may not be as important in their distribution and volume than low altitude (see main comment) L42: not true, you can assess the effect of changing microclimate not using altitudinal climate as you can be using different sampling time (which you did not do), different exposition, open top chambers etc. So, just focus on altitudinal but you can't say that other studies can't assess change of microclimate. L47: are the rang of altitude comparable and are the ecosystems comparable? L51: you cite articles on complete different ecosystems, such as Fierer et al 2011 (tropical), Meng et al 2013 (forest)... Focus on the Arctic and no other biomes, and same ecosystems (i.e. tundra and not forest) as you will not expect to have the same trends. Clearly state the location and ecosystems the studies you cited are based on. L59: this is not true. Your study is also true at your sites and at other sites the effects will differ. The number of studies help us to determine the common drivers across different sites. Your study is not better than others at that level. For generalisation, you could have cited Chu et al 2010 as global study of microbial diversity across the Arctic (Soil bacterial diversity in the Arctic is not fundamentally

different from that found in other biomes) L60: "Fundamental" this is a strong word, please rephrase

Materials and Methods L80-84: any idea of the percentage plant cover? Did you measure it? You don't mention mosses and lichens, but they represent a large part of the plant cover in Arctic tundra and are completely missing from your description. L85: you speak about the bedrock in the entire article but you never define/describe it. Could you give some information on it. L86: was an organic horizon present in the low altitude soils? L93: "kept frozen" at which temperature? L122: section 2.4 there are no references and no justification of your measurements choices: temperature, duration of incubation...? L131: why did you adjust the amount of soil based on TOC and in which way? Could this bring a bias if you have more soil for example in high altitude to compare you results? L154, 156: change "mL" to "ml" L161: I guess you checked also for homoscedasticity? L161: state clearly if you transformed or not the PLFA data. L163-164: the horizontal and vertical transect are both gradient (one vertical one horizontal). See comment at the beginning. L166: how was the forward selection done? L167: why did you use only P values adjusted by Holms corrections. Any reference for that? L172-173: it is really confusing when you speak about whole-plots vs splits-plots when you don't have a plot experiment. I am not sure what you refer to here. L176: what type of correlation did you use? Why there is no direct reference to correlation in the previous sentence?

Results L190-192: this is a repetition of L 204. L192, it is also wrong what you say as the low altitude site show higher soil moisture than high altitude. Delete the sentence. L187, 203: which gradient are you talking about, be clear. L214: cite the Table you refer to L217-218: say if the correlation is positive or negative when you mention correlation even if it is given in brackets. L220: finish the sentence by "while increased in Gr2 and Gr3". L223-225: this should be given in the materials and methods and justify why you should use it. This problematic for me and can bias your results as mentioned in the main comment. L233: what is the "whole PLFA profile"? You did not use all

the biomarkers in previous tests, it is not the same than MCS? L229-237: should cite figure 5? For example, L230 which figure you refer to? L250: change "typical" by "characterized"

Discussion L256: do you mean "did not" or "did" correspond. Looking at your plot, you have the same trend between soil and atmospheric, just few degrees' differences. Nothing surprising here. Your explanation is not logical, snow will insulate the soil from air temperature, so having less snow should make the soil temperature more similar to air temperature but at the beginning you say they don't correspond. So, what are you trying to say? L255-264: this whole section I am not sure what message you are trying to deliver. Nothing is really new here, and could be condensed in a shorter section L281: OM does not grow but increase. Change "growing" by "increasing" L285: "documented growing contribution of ", rephrase this is difficult English to understand L288-290: ok there is no vascular plant, but what about mosses and lichens? Could they be partly responsible for presence of sitosterol? You have to discuss about mosses and lichens in the article, you completely omitted to mentioned them, and I don't think they are not present on the soil. Lichens have a distribution up to high-altitude and you often find them on Svalbard on top of mountain even without any soil. L306: this reference is a bit old, is there any more recent references done on a larger number of bacteria? L307: what is the parental material? L309-314: of course if you divide microbial biomass and activity by TOC you will find an altitude effect, because there is less plant input at higher altitude so lower TOC. It does not mean your microbial biomass is higher at high altitude or there is higher microbial activity. Is it really important or interesting to know that there is higher proportion of living microbial biomass per soil TOC content? Your site effect if stronger on microbial biomass (PLFA /g soil) than altitude (only present for GR1) which is an interesting result on its own and show the variability of the suppose vertical gradient. L316-324: This is repetition of L309-324. You need to merge both section and make it shorter, and again that you have higher microbial biomass per TOC is not of a major interest. L330-331: keep in mind that you work on alkaline soil and it might be difficult to compare to other soil which are acidic. . . L348:

you are not working on "dynamics" because you only have one sampling time. Remove dynamic from the title as you can only make some assumption L349: Why do you think it is low quality litter? Compare to what? The fact you have a high C/N does not mean low quality but just more OM and less degraded. Low OM quality usually have low C/N. This also could contradict what you say L354 "which released easily assimilable". Is low quality litter easily assimilable? L352: So what? Ok it reduces the microbial biomass per unit of C, it just means that there is more C in the soil. You describe this as an issue but I don't see why it should be an issue? The microbial biomass does not decrease with increasing altitude! Sorry to repeat myself but you use the results your prefer to support your theory without considering your entire results. L356-357: this is a strange wording which make the sentence difficult to understand. What do you mean by "inverse consequences from soil MCS compared to development of microbial communities"? In which way this is "inverse" and how do you have an inverse microbial community structure (or just different) and why you speak about development or young soil when you don't measure microbial growth or dynamic and the age of the soil? L358: Are you talking of your study or not? You are not working on a "succession". You are again making too many assumption and extrapolation based on your results. This is also contradictory to what you say in the previous sentence. Here you say at the "maximal plant biomass" the fungi dominate, while you say bacteria dominate in the previous section and in general in the article. L360: so, any conclusion? Can you really make this comparison based on 1 study? L362-376: your results do not support what you say. You have one sampling time point, don't make assumption on what you don't measure: dynamic. Focus on your results. You can say that the presence of ergosterol coincide with continuous dominance of fungi at high altitude sites, but you don't need an entire section about it. Delete most of this section into one or few sentences

Conclusion: L379: move "were" just before "characterized" L380: this is not true. Unless you divide by TOC, there is no consistent effect of altitude on biomass and activity. L381: can you really say that there is negligible effect of microclimatic conditions over the summer with only one date of sampling? Do you think you have enough resolution

with your sampling strategy to assess the effect of summer microclimate? L382: again, you use gradient without saying which gradient you are talking about and when you define in the material and methods "gradient" to refer to horizontal not vertical. L383: you need to clearly state the decrease in pH. The decrease is less than a pH unit and the soil remains slightly alkaline. This is important because your results are likely to be completely different on acidic soil. L384-385: again, what is the bedrock at your sampling site? L386-388: well, there is plenty of unvegetated area at low altitude and even when there are plants. Your thinking must be developed to unvegetated area not only at high altitude. Do plants will colonise high altitude soil which are only few cm thick with global warming? Also give a reference for the potential increase in plant cover in the Arctic as several articles were recently published. L389: you can't really say that it diminishes the variability because it depends on plant species colonizing new area, the bedrock as you say. You don't measure variability with PLFA, the resolution in the method you use is not high enough. L390: you don't measure microbial diversity, how do you know this could have a negative effect? L393: again not true, you don't have a considerable microbial biomass and your measure of microbial activity is questionable. You just can't make this conclusion L379-394: there is no mention of the site effect even if you clearly have a site effect on microbial biomass and activity. This should be clearly stated as the vertical gradient is directly affect by the horizontal one in relation (in your study) to bedrock

———————————————————

---

## Referee Comment (RC2) · Anonymous Referee #2 · 17 Aug 2017

General Comments

The study by Kotas et al. was focused on changes in microbial biomass, activity, and broad community structure (based on PFLA) along altitudinal gradients in the Artic. This question has great significance concerning the implications of global warming on these ecosystems. The study consists of 3 different transects represented by 4 different elevations, and for each sample the authors collected substantial amounts of data representing soil type, soil chemistry (pH, ion content and concentrations, TOC, TN, moisture content, and temperature ranges), and very briefly mention vegetation coverage. The authors try to disentangle the impacts of all these along with elevation

on microbes using partial redundancy analysis as well as several other statistical approaches. They have a robust sample design with good replication to try and address this question.

I did have several issues with the manuscript. First, I found it very confusing that the authors kept referring to two different gradients, altitudinal (the main gradient of interest), and horizontal. However, this horizontal aspect is never discussed in the methods section and I assume it is referring to the south to north orientation of the 3 transects along the Petunia Bay. This needs to be clarified explicitly and its significance needs to be discussed. Is it expected there is a strong S-N effect? I assumed these 3 gradients were expected to be replicates of each other, but they have strong differences in soil characteristics and microbial community (particularly Gr1). This becomes more apparent in the Discussion, but the author's need to make this clear early on.

I also had concerns with their microbial respiration data and the authors need to justify their choice of a 2 week pre-incubation at 6 C. The pre-incubation will burn off all the labile carbon and drastically alters this respiration rate. This needs discussed as it can substantially alter the conclusions of a large portion of the paper.

The discussion is too long and wordy. I found it difficult to understand the main points the authors were trying to convey. It seemed to be rushed relative to the excellent writing of the rest of the manuscript and has multiple grammar issues. I also think that there was too much superfluous material that distracts from the main message. The authors spend a great deal of time discussing impacts due to plant biomass, but have no data presented quantitatively examining plant communities, biomass, root biomass, etc... A lot of this can be safely removed, especially in sections 4.1 and 4.4, as the degree of detail discussed doesn't add too much to the broader implications of the study.

With some mostly editorial changes focusing on clarifying the findings I think this paper represents a significant contribution towards Arctic research and understanding the

environmental parameters shaping microbial communities in this sensitive area.

Specific Comments

L124: I was interested in why the authors decided to pre-incubate the soils at 6 C (far above the mean of -3.8C, and below the max of 16.2, as well as different from the 5 C cut-off used in L186])? Also, why did the authors choose to pre-incubate for 2 weeks at this temperature? Is this typical for these kinds of measurements? I would think you want to minimize the pre-incubation time to prevent a strong bottle effect, as well as removing all your labile carbon.

L126: Is the specific respiration ratio typical to compare with the field? Is it possible to convert PLFA to a more generalizable unit (such as per cell, per g biomass etc...) using conversion factors?

L144: Is there a reference to support this sum? Are you not overcounting the bacterial contribution by summing general bacterial biomarkers with specific bacterial group biomarkers (Actinos, G-, G+)? Would it not be preferable to us general fungal : general bacterial only?

L189: Maybe change "In contrary" to "In contrast".

L214: Maybe add at the end "and was instead transect specific". I realize this is implied, but I feel it makes it clearer.

L213 – L227: This section is confusing to me. It is very surprising that microbial activity (as you assayed it) is not related to carbon or nitrogen content and is instead related to positively with Ca and negatively with Mg. I worry the trend in increasing respiration with altitude is due to the pre-incubation.

L228: Write out "Microbial Community Structure" in the header of this section.

L229: Gradient here is the transect? Does this mean there is a continuous change along the S-N transects or that each is different?

L230: Nice to see so much explained due to altitude!

L231: Which gradients? Elevation or between the transects? Please fix or clarify this terminology!

L229 – L233: These few sentences are quite confusing and I think readers would be helped if you clarify. If I understand, the microbial community structure is impacted by elevation, but even more so by how the soils change with elevation? You ran multiple different tests to parse out these effects at different levels? Also, is microbial community structure here a relative score or absolute values?

L237: Re-running the analysis with the selected variables was non-significant? Can you clarify this statement? Why do you want to run the forward selection if the variables selected do not significantly explain the microbial community composition? Is the main message of this part, that these variables are not significant while altitude is?

L240 – L251: Nice results! I think this is more interesting that the previous paragraph. However, there are a lot of grammar mistakes here, some listed below. Maybe re-write this section for clarity.

L243: missing a space

L247: "A similarly significant trend"

L248: Change to PFLAs.

L249: change discrepant to disparate

L249: Consider re-writing, this is a very long sentence that can be shortened, maybe "The most disparate site in terms of MCS was the highest elevation sampled along Gr1. It was typified by a high abundance of PLFAs specific to Actinobacteria and a lower abundance of fungal PFLAs compared to analogous sites along Gr2 and Gr3."

L255: What does this sentence mean?

L265: "positive surface energy balance had a strong..."

L273: This is an incredibly important but difficult to decipher sentence. I think a lot of the sentences above it can be shortened or removed, but this should be clarified. Do you mean that "Mean temperatures and temperature stability did not change with altitude in this study"? [Therefore, variations in your parameters due to altitude are not simply due to temperature differences?] Here I would start off with a stronger statement of what you mean, and then offer your support.

L277: Extremely important to clarify what gradient you are talking about here.

L277: Are you missing a "not". This is a confusing sentence.

L281 – L296: Simplify this! It is too wordy and difficult to follow. E.G. "We explain this discrepancy by the proximity of glacier stream, which could wash away the upper soil organic layer during abnormal spring-melt events in the past", can be changed to "The only exception was the lowest site of Gr2 which had similar OM content to higher elevation sites along the other transects. This is likely due to the proximity of a glacier stream, which would wash away the topsoil during a flood."

L284: "vascular plants also influenced"

L286: Please provide a citation for this.

L288 – L290: Is this important for your findings?

L290: Lots of grammar issues.

L292: Or high lichen components at high elevation?

L298 – L314: You need to discuss the implications of your pre-incubation step in this section. It can also be clarified or simplified for the readers.

L304-L308: Please include relevant concentrations of the Mg inhibitory effect here.

L309 – L314: This is a nice summary. However, the normalized characteristics are

inherently dependent on the soil OM, so isn't their increase directly due to the OM decrease?

L323 – L324: Please clarify this statement. What shift in resources lead to the slow accumulation of low quality OM? What are the ramifications of your pre-incubation when you are suggesting some samples are enriched in more recalcitrant OM?

L327 – L336: A lot of speculation. Is all this necessary?

L337 – L347: Very speculative.

L384: "bedrock chemistry were recognized as the main factors. . ."

L387 – L388: A confusing sentence, consider revising.

Figure2: Consider moving either this figure, or Table1 to the supplemental information to shorten the main paper.

Figure 4: How much variation is there between altitude replicates? Maybe add a supplementary figure showing ellipsoids or individual sample points.

---

## Author Comment (AC1) · 12 Sep 2017

Dear Associate Editor and Reviewer, please find below a detailed response to all Reviewer #1 comments and questions regarding our manuscript.

Best regards,

Petr Kotas and co-authors

General comments The authors investigate the effect of horizontal (across a valley) and vertical (altitude) gradients on microbial community structure (PLFA), biomass and activity in High Arctic. They found that both gradient affect microbial parameters, with

shift in the dominance of bacteria and fungi related to the chemistry of the bedrock. The study target interesting question, is relevant for publication in Biogeosciences and is overall well done. My main criticisms are the method used to measure microbial activity (main issue), and too many assumptions made outside the variables measured, going beyond what the results can show.

The measure of microbial activity seems unrealistic. First, 2 mm soil was used which was frozen and thaw prior to incubation in the lab. So, the microbial community and soil endure 1 freeze-thaw cycle + sieving that will affect OM availability and the microbial community. Then you left the samples for 14 days at 6 degrees before measuring the $CO_2$ for 24h, which you define as "basal respiration". Fourteen days represent 1/4 of the summer (> 5 degrees) in the Arctic (low altitude) or even your entire summer for the high-altitude site (Table 1), this is a significant amount of time in the Artic. There is no justification and references used to explain why you made these choices. Overall, we can doubt that the high altitude produce high $CO_2$ emissions in in-situ conditions and we can ask the values of your results regarding microbial activity. You did not discuss at any time the limitations of such measurement. We can imagine that the microbial community adapted better or took longer to adapt to incubation condition in high altitude soil explaining the higher $CO_2$ emissions at 14 days. We can also imagine that at low altitude, because of the higher TOC, the $CO_2$ emissions are high rapidly after thawing and after 14 there is not much activity, while for high altitude it took longer to mineralize more complex OM. In other words, your results of microbial activity could be just the results of your incubation/sample preparation. You need to fully acknowledge this in the article, and avoid any conclusion stating that high altitude is a hot spot of microbial activity because your data can't fully support this. You need to be much more conscious about the microbial activity result. Have you measure $CO_2$ emissions over time?

Author response: We admit that it was inappropriate to call the measured respiration "basal". The characteristic we measured is rather the "potential respiratory activity".

We agree with the reviewer that our methodological approach should be explained better. Since we were not able to conduct the measurements on site in freshly collected soils, we had to choose the proper methodology how to store and transport the samples and how to measure microbial activity to get a representative characteristic of the sites. The methodology was chosen based on available knowledge and our experiences with similar experiments. It was shown that refrigeration has stronger effect on microbial activity than freezing (Stenberg et al. 1998, SBB, 30, 393-402). It is likely because microbial activity does not stop at 4°C, which could lead to exhaustion of available substrates during the storage. Slow drying (another alternative for sample storage) followed by rewetting affects the microbial respiration similarly to freezing-thawing (Clein and Schimel, 1994, SBB, 26, 403-406) but it is far from conditions, which the soils face in the Arctic. We are aware of the responses of soil microbes to freezing-thawing cycles. Therefore, we measured soil microbial respiration repeatedly after 4, 12 and 14 days of the incubation at 6 °C (see Fig. 1). The expected respiratory burst occurred in all the samples during first 4 days, similarly as reported elsewhere (Skogland et al. 1998, Soil Ecol. 11, 147-160). It was estimated that up to 50% of the microbial biomass is killed following a single freeze-thaw cycle (Soulides and Allison, 1961, Soil Sci. 91, 291-298), leading to 10-40 fold increase in dissolved sugars and amino acids (Ivarson and Sowden 1966, 1970, Soil Sci. 46, 115-120 and 50, 191-198, respectively). After this $CO_2$ flush, the $CO_2$ production rate decreased and the mean respiration rates measured between days 4-12 and 12-14 did not already differ from each other. This pointed to a stabilization of microbial activities in the soils, as reported by Schimel and Clein (1996, SBB 28, 1061-1066). The respiration burst between days 0-4 were positively correlated with the respiration rates measured later (r=0.93 and 0.74, both P<0.0001, n=36). Therefore, there was a consistent difference among soil microbial activities along vertical gradient during the whole incubation. The samples from high altitudes showed higher flush of $CO_2$ than soils from lower altitudes (except Gr1) as well as higher potential respiration rates after stabilization. The respiration data together with other data which we reported show that the idea that soils in high

altitudes contain more complex soil OM and that it would take longer time to start and increase microbial activity after freezing-thawing is not correct. Instead, microbial activity in these soils is triggered as rapidly as in soils from lower altitudes. In summary, we insist that the presented respiration data are not an artefact of our storage/preparation procedure. We do not think that sample storage/preparation procedure distorted the differences in potential microbial activities occurring along vertical gradient. Further, our data showed that the respiration likely stabilized earlier than after 14 days in all samples independent of altitude. Nevertheless, we chose to present the stabilized respiration rates, not biased by the respiratory burst following freezing-thawing. However, we have additional data and can add them to the revised manuscript with proper explanation.

The discussion and conclusions are too long and go far beyond what you can say based on your results. There are many sections you discuss about the dynamic of microbial community but you only did one sampling time. You can't make big conclusions about dynamic of the system, such as L362-376. You need to just briefly mention potential dynamic but don't go much further. Similarly, you speak a lot about the effect of plant cover even you did not measure any parameters to characterize the plant cover (you also forgot to mention anything about mosses and lichens despite their importance in the Arctic) such as above ground biomass, root biomass, percentage cover (did you properly assessed it?), diversity. You just described the main vascular plants. So your section 4.4, is simply too long and not fully supported with your data. This entire section could be reduced in few sentences and focus on presence/absence of plants and not linked to microbial dynamic.

Author response: We agree with the reviewer's concern that discussion is too long. We also admit that our discussion interpretations go in some cases behind the measured data. We will revise the discussion to make it more straightforward and concise. Regarding the primary producers, we have data about plant and lichenized soil crust percentage cover at the sampling sites. These data will be added to the result section

and discussed. However, we didn't assessed the plant diversity and biomass at the sampling sites.

On the other hand, your discussion lack of putting your results in perspective with the literature, for example other studies investigating microbial community in bare/unvegetated soil (there is several article on this in the Arctic). It would be interesting to see if F/B ratio is similar in unvegetated a low altitude compared to other study.

Author response: We extensively searched the WOS and used all relevant literature sources which we found. However, we can't exclude the possibility that some relevant publications were omitted. Based on reviewer's recommendations, we will search for relevant literature again.

Your main conclusion should be the absence of plant rather than altitude effect, especially if we consider that high altitude soil in the Arctic (especially High Arctic) is likely to be rare, and also extremely shallow (only few cm) and not a massive stock of C. So we can wonder of their importance?

Author response: We don't fully understand these remarks. Isn't the decreasing plant abundance due to the altitude effect? Please consider our sampling strategy (L 88-92). What connection is there with rareness of high altitude soils? We don't think that the high altitude soils doesn't deserve scientific interest (despite of their low C stocks and shallow soil profile), especially in context of proceeding global warming and future development of ecosystems in the Arctic. We would like to point out that the high elevation habitats form significant part of the non-glaciated landscape not only in Svalbard, but also in other parts of the northern circumpolar region.

Also you should more conclude on the site effect you have (Gr1) as site is clearly an effect on microbial biomass (consistently lower) and structure for the high altitude site.

Author response: Even though we discussed the site effect in section 4.3., we will focus

more on the spatial heterogeneity in the data.

Another point you miss in your discussion is the fact that the soil you study is always alkaline (are alkaline soils prevalent on Svalbard or acidic?). You need to discuss or conclude if your results would be similar on acidic soil in the Arctic and compare with the relevant literature as pH is a big driver of microbial community including in the Arctic. This is an important point to make and be more critical about your results.

Author response: We thank the reviewer for pointing out this problem. Soils on Svalbard are neutral or alkaline. The soil pH in other parts of the northern circumpolar region is more variable, ranging from acidic pH (mainly on granite and gneiss bedrock) to highly alkaline pH. We are aware of pH effect on microorganisms. This issue will be discussed more thoroughly.

Finally, you find an effect of altitude on microbial biomass mainly when you divide the data by TOC. This bias the data and don't reveal hidden effect of altitude. Yes, you have less C at high altitude but you don't have more microbial biomass. The fact there is more biomass per unit of C is not of a major interest. Dividing your results by TOC is not important and bias your results. Focus on the altitude effect on microbial community structure and ratio, and acknowledge that there is not a major effect of altitude on biomass but rather a site effect.

Author response: We agree with the reviewer that we should emphasize the spatial variability in the microbial biomass and activity data (not normalized to soil TOC content) and not focus mainly on the normalized data. However, we don't fully agree with the reviewer's opinion that normalization bias the data. We didn't write that we have more microbial biomass at high altitude. Many papers focused on altitudinal gradients or polar areas were published based on microbial data normalized per TOC content (for activity see e.g. Schimel and Clein 1996 SBB 28, 1061-1066, Väre et al, 1997, Arctic and Alpine Res. 29, 93-104; for biomass e.g. Allison et al. 2007, SBB 39, 505-516, Xu et al. 2014, European Journal of Soil Biology, 64, 6-14; Djukic et al.

2010, SBB 42, 155-161, Väre et al, 1997, Arctic and Alpine Res. 29, 93-104). Author response: The reason for that was considering the differences in soil TOC content between sites. We did the same here as the microbial biomass is usually well correlated with soil TOC content (Wardle 1992, Biological reviews 67, 321-358). Even though there was site effect on microbial biomass and activity (again, we have to stress this in the manuscript), the normalized data show relatively uniform trends of low microbial biomass and activity in soils with highest C stocks and higher microbial biomass and activity with decreasing soil TOC content and increasing elevation (except for the most elevated sites along Gr1). The higher proportion of microbial C within total soil organic C further points to higher lability of the OM, which corresponds well with the high flushes of $CO_2$ from the soils as a response to freezing-thawing. Moreover, the altitudinal trends in microbial biomass and respiration did not always follow the altitudinal trends in TOC content (compare data in Table 3 with Fig. 3). For instance, we found the most pronounced decrease of TOC content with elevation along Gr1, but the microbial characteristics normalized per TOC content did not correspond to this trend. In contrast, the TOC content from the lowest and highest sites along Gr2 did not differ, but the altitudinal trend in microbial characteristics was significant. We consider this information as important characteristic of particular sampling sites.

Specific comments Altitude and transect are both gradient and not only transect. This is really confusing in the text when you speak about gradient, as it is unclear if you speak about vertical or horizontal. You can't refer to the horizontal gradient as "gradient" and the vertical one as altitude. Decide if you speak about vertical or horizontal gradient, or altitude and transect. Change in the entire text, but don't go from "gradient" for horizontal and then use gradient also for "vertical". Be consistent.

Author response: We agree with the reviewer opinion that this needs to be clarified throughout the text. We will clearly distinguish between the effects of altitude (vertical) and transect (horizontal) in the revised manuscript.

Introduction L20: true but it is simply related to less C from a plant origin, it does not

mean that there is more biomass. This is not true as well for GR1. You are telling only one part of the story here.

Author response: We didn't say that there was more microbial biomass at high altitude. However, we agree with the reviewer that the site effect on microbial biomass and activity (not normalized per TOC content) must be emphasized.

L21: "the 2 dominant microbial groups" it sounds like it is unusual or a result on its own but in the same time with PLFA you have access to fungi and bacteria only. Just speak about fungi and bacteria.

Author response: We agree with this comment. Sentence will be revised.

L23: you didn't measure microbial dynamic over time, only the change in soil temperature. So, I would focus on what you measured and not make assumption, especially in the abstract. Keep this for the discussion

Author response: Assumption will be removed from the abstract.

L25-26: the conclusion is an overstatement. In general, unvegetated area should be considered as previous studies showed. Speaking about high elevation as hotspots of microbial activity based on 1 measurement is an overstatement (see main comment).

Author response: Conclusions will be revised.

L36-41: this is normal as high altitude usually have no soil present or are extremely shallow (few cm) and may not be as important in their distribution and volume than low altitude (see main comment).

Author response: We completely agree with the reviewer that the high altitude soils are more important in distribution and volume compared to high elevation soils. However, we wanted to point out in these lines that the soil microbial properties were not thoroughly studied yet along the altitudinal gradients in the Arctic. We consider the information given in these lines relevant.

L42: not true, you can assess the effect of changing microclimate not using altitudinal climate as you can be using different sampling time (which you did not do), different exposition, open top chambers etc. So, just focus on altitudinal but you can't say that other studies can't assess change of microclimate.

Author response: The sentence will be revised according to reviewer's comment.

L47: are the ranges of altitude comparable and are the ecosystems comparable?

Author response: No, they are not. We are writing here about general altitudinal trends. The referred studies investigated microbial communities in different latitudes and across different altitudinal ranges (please see our next response).

L51: you cite articles on complete different ecosystems, such as Fierer et al 2011 (tropical), Meng et al 2013 (forest)... Focus on the Arctic and no other biomes, and same ecosystems (i.e. tundra and not forest) as you will not expect to have the same trends. Clearly state the location and ecosystems the studies you cited are based on.

Author response: We are aware of that. General latitudinal, but also altitudinal trends in diversity of animals and plants are one of the most widely recognized patterns in ecology. However, they are not valid for the microbial diversity as we wanted to show here. The number of references about microbial diversity or community structure along altitudinal gradients from the Arctic is strongly limited. We will mention the ecosystems the studies we cited are based on.

L59: this is not true. Your study is also true at your sites and at other sites the effects will differ. The number of studies help us to determine the common drivers across different sites. Your study is not better than others at that level. For generalisation, you could have cited Chu et al 2010 as global study of microbial diversity across the Arctic (Soil bacterial diversity in the Arctic is not fundamentally different from that found in other biomes)

Author response: We completely agree with the reviewer opinion – our study is not

better than others and the effect is site specific, which does not allow generalization – as we have written here. What is not true then? We didn't want to highlight our study here.

L60: "Fundamental" this is a strong word, please rephrase.

Author response: Sentence will be rephrased.

Materials and Methods L80-84: any idea of the percentage plant cover? Did you measure it? You don't mention mosses and lichens, but they represent a large part of the plant cover in Arctic tundra and are completely missing from your description.

Author response: Regarding the primary producers, we have data about plant and lichenized soil crust percentage cover at the sampling sites. These data will be added to the result section and discussed. However, we didn't assessed the plant diversity and biomass at the sampling sites.

L85: you speak about the bedrock in the entire article but you never define/describe it. Could you give some information on it.

Author response: We would like to thank the reviewer for pointing out this deficit. We have detailed information about geology of the Petunia Bay. The information will be added to the site description.

L86: was an organic horizon present in the low altitude soils?

Author response: No, the soil profile is poorly developed. Based on our experience there is rather litter layer on the soil surface and then relatively homogeneous mineral soil layer overlaying coarse gravel.

L93: "kept frozen" at which temperature?

Author response: At -20 °C

L122: section 2.4 there are no references and no justification of your measurements

choices: temperature, duration of incubation. . .?

Author response: The information will be added. The incubation temperature of 6 °C was chosen as it represents mean summer soil temperature across the whole elevational gradient (mean summer temperatures for particular elevational levels ranged from 5.3 to 7.1, see Table 1; the mean summer temperature along the whole altitude range was 6 °C). Please find the justification of our measurement choices in our response to general comments.

L131: why did you adjust the amount of soil based on TOC and in which way? Could this bring a bias if you have more soil for example in high altitude to compare you results?

Author response: We optimized the PLFA extraction protocol in our laboratory to suit wide range of different soil types (with respect to the amount of lipids and size of the SPE cartridge for lipid fractionation, analytical procedure and stock sample aliquots for eventual reanalysis). As we use 0.7-1g of soil with TOC content around 5% and the microbial biomass is usually proportional to TOC content, we adjust the sample size for C "poor" soils. This modification could not bias our results since the extraction efficiency is not affected and the PLFA yield is thus comparable for all samples. Rather the opposite is true – not accounting for low TOC content could lead to concentrations of particular PLFAs below the detection limit.

L154, 156: change "mL" to "ml"

Author response: The abbreviation mL was used in all recent articles published in BGS.

L161: I guess you checked also for homoscedasticity?

Author response: Yes, we checked the data also for homoscedasticity. We will mention this in the Materials and Methods section.

L161: state clearly if you transformed or not the PLFA data.

Author response: The relative data were long-transformed. We will mention this in the Materials and Methods section.

L163-164: the horizontal and vertical transect are both gradient (one vertical one horizontal). See comment at the beginning.

Author response: We agree with the reviewer. We will clearly distinguish between the effects of altitude and transect in the revised manuscript.

L166: how was the forward selection done?

Author response: It was performed using CANOCO 5.0 software. The soil geochemical parameters were used as explanatory variables while the relative abundances of microbial groups (MCS) were used as the dependent variables (RDA). The test offer list of candidate variables sorted according to their contribution to total explained variation in the dependent variables, together with their significance. After selection of the best candidate, the contribution and significance of remaining candidates is recalculated to explain the remaining variability in the dependent data. Then the next candidate can be selected (of course only if significant).

L167: why did you use only P values adjusted by Holms corrections. Any reference for that?

Author response: We used the significant values adjustment to reflect the multiple tests performed on the same dataset. The Holm's correction follows the approach described in Holm (1979): A simple sequentially rejective multiple test procedure. Scand. J. Stat. 6: 65-70. This procedure is slightly less conservative compared to the often recommended Bonferroni correction. On the other hand, it is a sequential procedure and takes into account that the candidate predictors with stronger effect were selected first. Thus it suits better for the forward selection procedure (please see above our comment on the forward selection).

L172-173: it is really confusing when you speak about whole-plots vs splits-plots when

you don't have a plot experiment. I am not sure what you refer to here.

Author response: We described the permutation strategy in the constrained multivariate tests (RDA) here. As we mentioned in lines 171-172, we could assume that the characteristics of each sample will be autocorrelated with characteristics from other two samples taken from the same site (otherwise we had 9 independent transects and not three, which is not the case). In other words, the samples from the triplicate cannot be considered as independent samples due to relatively low inter-sample distance. The sampling design was in this context hierarchical with repeated measurements for each sampling site.

L176: what type of correlation did you use? Why there is no direct reference to correlation in the previous sentence?

Author response: We used correlations to find out how tightly were two variables related to each other. We will mention the correlations in the previous sentence.

Results L190-192: this is a repetition of L 204. L192, it is also wrong what you say as the low altitude site show higher soil moisture than high altitude. Delete the sentence.

Author response: Sentence will be deleted.

L187, 203: which gradient are you talking about, be clear. L214: cite the Table you refer to L217-218: say if the correlation is positive or negative when you mention correlation even if it is given in brackets. L220: finish the sentence by "while increased in Gr2 and Gr3".

Author response: The changes will be done according to reviewers comments.

L223-225: this should be given in the materials and methods and justify why you should use it. This problematic for me and can bias your results as mentioned in the main comment.

Author response: Please see or response to the reviewer's main comments.

[Figure]

L233: what is the "whole PLFA profile"? You did not use all the biomarkers in previous tests, it is not the same than MCS?

Author response: As we mentioned in L162-163, the MCS is the relative abundance of microbial groups (fungi, Gram-positive and Gram-negative bacteria etc.). We also performed the tests using the whole PLFA profile, ie. relative abundances of individual microbial PLFAs. We thank the reviewer for this comments, we must mention this in the section 2.7.

L229-237: should cite figure 5? For example, L230 which figure you refer to?

Author response: There is no figure showing the results commented on L229-231. We think that figures showing altitude effect (Fig. 5) and relation between selected environmental variables and MCS (Fig. 4) are more important. However, we would like to modify Fig. 4. by inclusion of sample points and envelopes to depict how much variation is between the altitude replicates and how distinct are the samples from particular transects. Please see the modified figure below.

L250: change "typical" by "characterized"

Author response: Sentence will be revised.

Discussion L256: do you mean "did not" or "did" correspond. Looking at your plot, you have the same trend between soil and atmospheric, just few degrees' differences. Nothing surprising here. Your explanation is not logical, snow will insulate the soil from air temperature, so having less snow should make the soil temperature more similar to air temperature but at the beginning you say they don't correspond. So, what are you trying to say?

Author response: We mean "did not" as written in line 256. Yes, the trends are similar. The main message here is about altitudinal stratification, not about trends. While the soil temperatures are stratified according to altitude (Fig. 1 and S1a), the air temperatures are not (Fig. S1b). The latter is not surprising, but different mean winter soil

temperatures along the altitudinal gradient, ranging from -4 to -10 °C between the least and the most elevated sites, can have strong implications for microbial activity (please see e.g. Drotz et al. 2010, PNAS 107, 201046-21051, and references therein). We also see logic in our explanation – more snow at lower elevation insulates the soil. Consequently, the difference between soil and air temperatures is much higher at low elevations compared to more elevated sites, where the soil and air temperatures correspond much better.

L255-264: this whole section I am not sure what message you are trying to deliver. Nothing is really new here, and could be condensed in a shorter section.

Author response: We will clarify and shorten our statements in the first paragraph of section 4.1.

L281: OM does not grow but increase. Change "growing" by "increasing" L285: "documented growing contribution of ", rephrase this is difficult English to understand

Author response: Sentences will be rephrased.

L288- 290: ok there is no vascular plant, but what about mosses and lichens? Could they be partly responsible for presence of sitosterol? You have to discuss about mosses and lichens in the article, you completely omitted to mentioned them, and I don't think they are not present on the soil. Lichens have a distribution up to high-altitude and you often find them on Svalbard on top of mountain even without any soil.

Author response: We agree with the reviewer's comment that the importance of lichens must be thoroughly discussed.

L306: this reference is a bit old, is there any more recent references done on a larger number of bacteria?

Author response: Unfortunately not, we didn't find any other relevant publications.

L307: what is the parental material?

Author response: It refers to bedrock. Will be changed.

L309-314: of course if you divide microbial biomass and activity by TOC you will find an altitude effect, because there is less plant input at higher altitude so lower TOC. It does not mean your microbial biomass is higher at high altitude or there is higher microbial activity. Is it really important or interesting to know that there is higher proportion of living microbial biomass per soil TOC content? Your site effect if stronger on microbial biomass (PLFA /g soil) than altitude (only present for GR1) which is an interesting result on its own and show the variability of the suppose vertical gradient.

Author response: Even though we agree with the reviewer that the spatial variability in the microbial biomass and activity data (not normalized to soil TOC content) should be emphasized, we consider the microbial characteristics normalized to TOC content important indicator of functioning of the soil system. If the microbial biomass was uniformly proportional to TOC content, the normalized microbial characteristics didn't show any altitudinal trend. We didn't say that total microbial biomass or activity is higher at higher elevations. Also, we don't think that the trends in normalized microbial variables are not interesting and important as they point to differences in a lability of organic matter and C sequestration. We consider this information as important characteristic of particular sampling sites.

L316-324: This is repetition of L309-324. You need to merge both section and make it shorter, and again that you have higher microbial biomass per TOC is not of a major interest.

Author response: We will merge both sections.

L330-331: keep in mind that you work on alkaline soil and it might be difficult to compare to other soil which are acidic...L348: you are not working on "dynamics" because you only have one sampling time. Remove dynamic from the title as you can only make some assumption

Author response: We completely agree with the reviewer.

L349: Why do you think it is low quality litter? Compare to what? The fact you have a high C/N does not mean low quality but just more OM and less degraded. Low OM quality usually have low C/N. This also could contradict what you say L354 "which released easily assimilable". Is low quality litter easily assimilable?

Author response: Low quality litter is not easily assimilable, but we referred in L354 to exudates and they are easily assimilable. However, the exudation represents small fraction of plant OM input. We considered the shift in substrate origin from N poor and structural compounds rich plant detritus at low elevations to N-rich microbial products at high elevations as a reason for higher microbial activity at the most elevated sites. We definitely need to discuss the presence of lichens here.

L352: So what? Ok it reduces the microbial biomass per unit of C, it just means that there is more C in the soil. You describe this as an issue but I don't see why it should be an issue? The microbial biomass does not decrease with increasing altitude! Sorry to repeat myself but you use the results you prefer to support your theory without considering your entire results.

Author response: Again, we admit that we have to focus to spatial variability in not-normalized microbial biomass and activity.

L356-357: this is a strange wording which make the sentence difficult to understand. What do you mean by "inverse consequences from soil MCS compared to development of microbial communities"? In which way this is "inverse" and how do you have an inverse microbial community structure (or just different) and why you speak about development or young soil when you don't measure microbial growth or dynamic and the age of the soil?

Author response: The meaning of "inverse" is as follows: while during succession is the increasing plant productivity accompanied by increasing fungal abundance in the

microbial communities, the fungal abundance along the elevational gradients increases together with decreasing plant occurrence. We didn't compare here the age of the soils, but successional development versus altitudinal climosequence – it has very similar attributes, e.g. gradient in plant occurrence and productivity, organic matter content etc. That is why we compared the F/B ratios in lines 359-361. The sentence will be clarified.

L358: Are you talking of your study or not? You are not working on a "succession". You are again making too many assumption and extrapolation based on your results. This is also contradictory to what you say in the previous sentence. Here you say at the "maximal plant biomass" the fungi dominate, while you say bacteria dominate in the previous section and in general in the article.

Author response: We are not talking about our study here. We are talking about the contrasting F/B ratios in early successional stages of soil development and soil at the most elevated sites. Despite the fact that both have low TOC contents and low plant biomass, the F/B ratio strongly differs.

L360: so, any conclusion? Can you really make this comparison based on 1 study? L362-376: your results do not support what you say. You have one sampling time point, don't make assumption on what you don't measure: dynamic. Focus on your results. You can say that the presence of ergosterol coincide with continuous dominance of fungi at high altitude sites, but you don't need an entire section about it. Delete most of this section into one or few sentences.

Author response: We agree with the reviewer opinion that this section must be significantly shortened.

Conclusion: L379: move "were" just before "characterized" L380: this is not true. Unless you divide by TOC, there is no consistent effect of altitude on biomass and activity. L381: can you really say that there is negligible effect of microclimatic conditions over the summer with only one date of sampling? Do you think you have enough resolution

with your sampling strategy to assess the effect of summer microclimate? L382: again, you use gradient without saying which gradient you are talking about and when you define in the material and methods "gradient" to refer to horizontal not vertical. L383: you need to clearly state the decrease in pH. The decrease is less than a pH unit and the soil remains slightly alkaline. This is important because your results are likely to be completely different on acidic soil. L384-385: again, what is the bedrock at your sampling site? L386-388: well, there is plenty of unvegetated area at low altitude and even when there are plants. Your thinking must be developed to unvegetated area not only at high altitude. Do plants will colonise high altitude soil which are only few cm thick with global warming? Also give a reference for the potential increase in plant cover in the Arctic as several articles were recently published. L389: you can't really say that it diminishes the variability because it depends on plant species colonizing new area, the bedrock as you say. You don't measure variability with PLFA, the resolution in the method you use is not high enough. L390: you don't measure microbial diversity, how do you know this could have a negative effect? L393: again not true, you don't have a considerable microbial biomass and your measure of microbial activity is questionable. You just can't make this conclusion L379-394: there is no mention of the site effect even if you clearly have a site effect on microbial biomass and activity. This should be clearly stated as the vertical gradient is directly affect by the horizontal one in relation (in your study) to bedrock.

Author response: The conclusions will be revised.

———————————————

[Figure]

[Figure]

Fig. 1 Comparison of mean daily $CO_2$ production at days 0-4, 4-12 and 13-14 (respiration presented in the manuscript).

[Figure]

Revised Fig. 4

---

## Author Comment (AC2) · 12 Sep 2017

Dear Associate Editor and Reviewer, please find below a detailed response to all Reviewer #1 comments and questions regarding our manuscript.

Best regards,

Petr Kotas and co-authors

General comments: The study by Kotas et al. was focused on changes in microbial biomass, activity, and broad community structure (based on PFLA) along altitudinal gradients in the Artic. This question has great significance concerning the implications

of global warming on these ecosystems. The study consists of 3 different transects represented by 4 different elevations, and for each sample the authors collected substantial amounts of data representing soil type, soil chemistry (pH, ion content and concentrations, TOC, TN, moisture content, and temperature ranges), and very briefly mention vegetation coverage. The authors try to disentangle the impacts of all these along with elevation on microbes using partial redundancy analysis as well as several other statistical approaches. They have a robust sample design with good replication to try and address this question.

I did have several issues with the manuscript. First, I found it very confusing that the authors kept referring to two different gradients, altitudinal (the main gradient of interest), and horizontal. However, this horizontal aspect is never discussed in the methods section and I assume it is referring to the south to north orientation of the 3 transects along the Petunia Bay. This needs to be clarified explicitly and its significance needs to be discussed. Is it expected there is a strong S-N effect? I assumed these 3 gradients were expected to be replicates of each other, but they have strong differences in soil characteristics and microbial community (particularly Gr1). This becomes more apparent in the Discussion, but the author's need to make this clear early on.

Author response: We agree with the reviewer opinion. Even though both the horizontal and altitudinal aspects were mentioned in the methods (L 163) and presented in the results (L197-199, L229-231), we agree that this needs to be clarified throughout the text with stronger attention paid also to horizontal variability. We will clearly distinguish between the effects of altitude (vertical aspect) and transect (horizontal aspect) in the revised manuscript. The 3 transects were expected to be replicates of each other. We didn't expect any variability in soil geochemical or microbial characteristics which could be ascribed to the differences in orientation of the selected transects. Opposite was true - we did our best to select similarly oriented transects (slopes on the western coast of Petunia Bay) in order to minimize the effect of distinct slope orientation.

I also had concerns with their microbial respiration data and the authors need to justify

their choice of a 2 week pre-incubation at 6 C. The pre-incubation will burn off all the labile carbon and drastically alters this respiration rate. This needs discussed as it can substantially alter the conclusions of a large portion of the paper.

Author response: We agree with the reviewer that we have to justify and discuss our methodological approach. The methodology was chosen according to available knowledge and our experiences with similar experiments. We insist that the presented respiration data corresponds to in situ microbial activity. First of all, we measured the respiration also at day 4 and 12 during the incubation period. Our measurements have shown two important things: i) the daily production of $CO_2$ during the first four days of incubation was on average 2.6 times higher compared to daily $CO_2$ production between days 4 and 12 (Fig. 1). This is accordance with strongly enhanced respiratory burst after soil thawing reported previously (Skogland et al. 1998, Soil Ecol. 11, 147-160) due to the flush of easily available substrates from lysed microbial cells. It was estimated that up to 50% of the microbial biomass is killed following a single freeze-thaw cycle (Soulides and Allison, 1961, Soil Sci. 91, 291-298), leading to 10-40 fold increase in dissolved sugars and amino acids (Ivarson and Sowden 1966, 1970, Soil Sci. 46, 115-120 and 50, 191-198, respectively). After this $CO_2$ flush, the $CO_2$ production rate decreased and the mean respiration rates measured between days 4-12 and 12-14 did not already differ from each other. This pointed to a stabilization of microbial activities in the soils, as reported by Schimel and Clein (1996, SBB 28, 1061-1066). Therefore we chose relatively long pre-incubation period to ensure that we measured the stabilized respiration not biased by the respiratory burst; ii) The respiration burst between days 0-4 were positively correlated with the respiration rates measured later (r=0.93 and 0.74, both P<0.0001, n=36). Therefore, there was a consistent difference among soil microbial activities along vertical gradient during the whole incubation. Even though the final respiration measurement could be conducted earlier, we are confident that measurement at day 14 did not significantly affect the trends or the absolute values of microbial respiration presented in the manuscript.

The discussion is too long and wordy. I found it difficult to understand the main points the authors were trying to convey. It seemed to be rushed relative to the excellent writing of the rest of the manuscript and has multiple grammar issues. I also think that there was too much superfluous material that distracts from the main message. The authors spend a great deal of time discussing impacts due to plant biomass, but have no data presented quantitatively examining plant communities, biomass, root biomass, etc. A lot of this can be safely removed, especially in sections 4.1 and 4.4, as the degree of detail discussed doesn't add too much to the broader implications of the study.

Author response: Again, we agree with these comments. Discussion will be revised and shortened, especially in section 4.1. Regarding section 4.4., we will provide the information about plant and lichenized soil crust percentage cover as these data are recently available for the sampling sites.

With some mostly editorial changes focusing on clarifying the findings I think this paper represents a significant contribution towards Arctic research and understanding the environmental parameters shaping microbial communities in this sensitive area.

Specific Comments L124: I was interested in why the authors decided to pre-incubate the soils at 6 C (far above the mean of -3.8C, and below the max of 16.2, as well as different from the 5 C cut-off used in L186])?

Author response: The incubation temperature of 6 °C was chosen as it represents mean summer soil temperature along the whole elevational gradient (mean summer temperatures for particular elevational levels ranged from 5.3 to 7.1, see Table 1; the mean summer temperature across the whole gradient is 6 °C).

Also, why did the authors choose to pre-incubate for 2 weeks at this temperature? Is this typical for these kinds of measurements? I would think you want to minimize the pre-incubation time to prevent a strong bottle effect, as well as removing all your labile carbon.

Author response: Please see our response to general comments about our respiration measurements above.

L126: Is the specific respiration ratio typical to compare with the field? Is it possible to convert PLFA to a more generalizable unit (such as per cell, per g biomass etc.) using conversion factors?

Author response: The specific respiration was used primarily to reveal the variability and general patterns in the microbial activity per unit of microbial biomass. In our view, the observed trends are the most important massage. Soil PLFA content is generally accepted as quantitative measure of microbial biomass. We don't think that conversion of soil PLFA content to microbial biomass carbon (or per cell) could add any value to the information presented in the manuscript. The conversion factors vary in the literature sources and are inevitably affected by cell morphology (membrane area versus cell biovolume). There is different PLFA to microbial biomass ratio not only for fungi and bacteria, but also for bacterial cells differing in size and shape. As the fungi to bacteria ratios varied significantly between sites, we consider any recalculation using a single conversion factor as speculative and hardly employable for comparison with other studies based on measurements of soil microbial carbon content (e.g. by chloroform fumigation method).

L144: Is there a reference to support this sum? Are you not overcounting the bacterial contribution by summing general bacterial biomarkers with specific bacterial group biomarkers (Actinos, G-, G+)? Would it not be preferable to us general fungal : general bacterial only?

Author response: The bacterial abundance is in majority (if not all) of papers using PLFA as quantitative measure of microbial biomass calculated as a sum of all markers specific to bacteria. The specific bacterial groups (Actinobacteria, G-, G+) belongs to bacteria and they need to be considered when calculating the F/B ratio. Considering only general bacterial markers, which are specific to bacteria but cannot be ascribed to

one of the above mentioned bacterial groups, would lead to significant overestimation of fungal presence in the soil (references e.g. Frostegård and Bååth 1996, Biol. Fert. Soils 22, 59-65; Bååth and Anderson 2003 SBB 35, 955-965; Kaiser et al. 2010, New Phytologist 187, 843-858).

L189: Maybe change "In contrary" to "In contrast".

Author response: We agree

L214: Maybe add at the end "and was instead transect specific". I realize this is implied, but I feel it makes it clearer.

Author response: We agree

L213 – L227: This section is confusing to me. It is very surprising that microbial activity (as you assayed it) is not related to carbon or nitrogen content and is instead related to positively with Ca and negatively with Mg. I worry the trend in increasing respiration with altitude is due to the pre-incubation.

Author response: This relationship between respiration and base cation availabilities was surprising also for us. However, the microbial activity (respiration in this case) doesn't have to correspond with biomass as was shown previously (Šantrůčková and Straškraba, 1991, SBB 23, 525-532). Moreover, available nutrients rather than total C and N stocks affect microbial activity (unfortunately, we were not able to extract the available nutrients in the field and this information is missing in our dataset). Based on the background data from our respiration measurements (please see above our response to general comments), we insist that the presented respiration data are not a result of our pre-incubation step and can be used as potential respiratory activity of soil microbes. We thus believe that soil geochemical properties such as high magnesium availability can be very important drivers of microbial activity and abundance in these arctic soils.

L228: Write out "Microbial Community Structure" in the header of this section.

Author response: We agree

L229: Gradient here is the transect? Does this mean there is a continuous change along the S-N transects or that each is different?

Author response: Yes, gradient is transect here. The results mean that there is a significant shift in the MCS not only between elevations, but also significant differences between transects in horizontal direction. We admit that the horizontal/vertical aspects must be commented more clearly throughout the manuscript.

L230: Nice to see so much explained due to altitude! L231: Which gradients? Elevation or between the transects? Please fix or clarify this terminology!

Author response: Terminology will be clarified throughout the manuscript.

L229 – L233: These few sentences are quite confusing and I think readers would be helped if you clarify. If I understand, the microbial community structure is impacted by elevation, but even more so by how the soils change with elevation? You ran multiple different tests to parse out these effects at different levels? Also, is microbial community structure here a relative score or absolute values?

Author response: We will clarify these statements. Let us to offer brief explanation: the microbial community structure significantly changed along the elevational gradients and between transects (ie. both factors, transect and elevation, were significant; L229-231). The significant effects of transect and elevation can be well explained by spatial variability in the soil geochemical properties which were determined (ie. horizontal and altitudinal variability in the soil properties, L 231-233). The MCS used here (and in general throughout the manuscript) are relative abundances of microbial groups (not scores, see L161-164 in Method section).

L237: Re-running the analysis with the selected variables was non-significant? Can you clarify this statement? Why do you want to run the forward selection if the variables selected do not significantly explain the microbial community composition? Is the main

message of this part, that these variables are not significant while altitude is?

Author response: As we mentioned in the previous comment, the effect of transect and elevation on microbial community structure could be explained by the variability in soil properties retained by forward selection (see also section 3.2. in the manuscript). However, we wanted to find out whether the retained soil properties sufficiently explain the elevation and transect effect (the explanatory variables never explain 100% of variability in the community composition). Thus we used the variables retained by forward selection (ie. variables with the highest power to explain variability in the MCS, see L165-166) as covariates (ie. we tested just the remaining variability in the MCS not associated with these variables), assuming that if there are missing important environmental variables that control the spatial variability in MCS, the test on transect and/or elevation effect will remain significant (L168-169). Only the elevation effect remained significant, meaning that the retained soil properties satisfactorily explained the differences between transects, but not the elevational trends. In other words, there are still missing some environmental variables in our dataset which shape the MCS along the elevational gradients. We consider this information interesting and important.

L240 – L251: Nice results! I think this is more interesting that the previous paragraph. However, there are a lot of grammar mistakes here, some listed below. Maybe re-write this section for clarity.

Author response: Will be clarified.

L243: missing a space L247: "A similarly significant trend" L248: Change to PFLAs. L249: change discrepant to disparate

Author response: Will be corrected.

L249: Consider re-writing, this is a very long sentence that can be shortened, maybe "The most disparate site in terms of MCS was the highest elevation sampled along Gr1. It was typified by a high abundance of PLFAs specific to Actinobacteria and a

lower abundance of fungal PFLAs compared to analogous sites along Gr2 and Gr3."

Author response: Will be rewritten.

L255: What does this sentence mean? Author response: The whole section 4.1. will be shortened, especially first two paragraphs. The sentence will be removed. L265: "positive surface energy balance had a strong.."

Author response: Will be rewritten.

L273: This is an incredibly important but difficult to decipher sentence. I think a lot of the sentences above it can be shortened or removed, but this should be clarified. Do you mean that "Mean temperatures and temperature stability did not change with altitude in this study"? [Therefore, variations in your parameters due to altitude are not simply due to temperature differences?] Here I would start off with a stronger statement of what you mean, and then offer your support.

Author response: We mean that mean temperatures and temperature stability (diurnal temperature fluctuation) does not change with elevation as we expected – ie. temperature will decrease with increasing elevation and the microclimate will be less stable in higher altitudes. We also expected generally higher fluctuation of soil moisture. However, we found very similar temperature conditions in the lowest and highest elevations, while the mid-elevated sites experienced warmer but less stable summer soil microclimate. The most important microclimatic parameter thus seemed to be the length of vegetation season and its effect on vegetation. We will rewrite this section in order to keep it concise.

L277: Extremely important to clarify what gradient you are talking about here.

Author response: This will be clarified throughout the manuscript. Please see also our response to your comment regarding L237.

L277: Are you missing a "not". This is a confusing sentence.

Author response: Corrected sentence: "...while the effect of transect was not significant".

L281 – L296: Simplify this! It is too wordy and difficult to follow. E.G. "We explain this discrepancy by the proximity of glacier stream, which could wash away the upper soil organic layer during abnormal spring-melt events in the past", can be changed to "The only exception was the lowest site of Gr2 which had similar OM content to higher elevation sites along the other transects. This is likely due to the proximity of a glacier stream, which would wash away the topsoil during a flood."

Author response: We agree that the paragraph is too wordy. Paragraph will be shortened.

L284: "vascular plants also influenced"

Author response: Sentence will be reworded.

L286: Please provide a citation for this.

Author response: Sinsabaugh et al. (1997). Reference given in Materials and Methods and will be provided in discussion.

L288 – L290: Is this important for your findings?

Author response: We explained the occurrence of $\beta$-sitosterol as indicator of plant derived organic matter transported from lower elevations.

L290: Lots of grammar issues. L292: Or high lichen components at high elevation?

Author response: We agree that the importance of lichens must be thoroughly discussed. However, lichens contain algal and cyanobacterial photobionts so there is not a conflict with our statement.

L298 – L314: You need to discuss the implications of your pre-incubation step in this section. It can also be clarified or simplified for the readers.

Author response: We will discuss the implications of our pre-incubation step. The whole paragraph will be revised.

L304-L308: Please include relevant concentrations of the Mg inhibitory effect here.

Author response: We would like to thank the reviewer for this comment. The inhibitory concentrations of $Mg^{2+}$ in solution were above 5 p.p.m and 50 p.p.m. for G- and G+ bacterial species, respectively (Webb 1949, Microbiology 3, 410‒424). The limiting concentrations will be mentioned in the discussion.

L309 – L314: This is a nice summary. However, the normalized characteristics are inherently dependent on the soil OM, so isn't their increase directly due to the OM decrease?

Author response: Not completely. The altitudinal trends in microbial biomass and respiration did not always follow the altitudinal trends in TOC content (compare data in Table 3 with Fig. 3). There was high variability in OC content along the particular transects. For instance Gr1 shown the most pronounced decrease of OC content with elevation, but the microbial characteristics normalized per OC content did not correspond to this trend. In contrast, the TOC content from the lowest and highest sites along Gr2 did not differ, but the altitudinal trend in microbial characteristics was significant. We thus don't agree with the opinion that use of microbial characteristics normalized per TOC content doesn't add any other information beside that there is a natural gradient in TOC content. Many papers were published based on microbial data normalized per TOC content only. We consider this information as important characteristic of particular sampling sites and indication of differences in a lability of organic matter and soil C sequestration.

L323 – L324: Please clarify this statement. What shift in resources lead to the slow accumulation of low quality OM? What are the ramifications of your pre-incubation when you are suggesting some samples are enriched in more recalcitrant OM?

Author response: We agree that this statement is dubious and confusing. The meaning is that high elevation habitats have higher proportion of active microbial biomass per OM content, including microbial primary producers (ie. microalgae; we admit that their presence in lichens must be discussed). Their necromass is much more vulnerable for decomposition compared to the plant litter. The higher productivity of plants and slow decomposition of their litter lead to TOC accumulation in the lower elevated soils, while the predominantly microbial primary production at the most elevated sites offer more available substrate for microbial growth. We consider this as the main reason for the observed pattern of high microbial activity per TOC content in the most elevated sites

L327 – L336: A lot of speculation. Is all this necessary?

Author response: We agree that this paragraph could be shortened. However, the F-B ratio is important indicator of microbial community composition and functioning. In our view is the interpretation of observed changes in F-B ratio and comparison with published data important.

L337 – L347: Very speculative.

Author response: We believe that Mg2+ availability is very important factor shaping MCS along the transects. It largely explained the trends in G-/G+ bacteria ratios (compare Table 3 and Fig. 6c, d in the manuscript). It was shown that growth of G- and G+ bacteria is limited at very different Mg2+ concentration levels (difference of one order of magnitude, see our response to comments on L304-308). The Mg2+ availability in the investigated soils exceeded these limiting concentrations, especially for G- bacteria (considering all available Mg2+ in soil solution and average soil moisture content 30%, the Mg2+ concentrations ranged approximately from 50-420 p.p.m.). We thus consider the given interpretation of observed shifts in MCS due to Mg2+ availability (Mg2+ availability was retained by RDA with forward selection of explanatory variables) as critical evaluation of relevant literature. However, we admit that statements about substitution of fungi by Actinobacteria are speculative and will be removed. The paragraph will be

shortened.

L384: "bedrock chemistry were recognized as the main factors"

Author response: Sentence will be reworded.

L387 – L388: A confusing sentence, consider revising.

Author response: Sentence will be revised.

Figure2: Consider moving either this figure, or Table1 to the supplemental information to shorten the main paper.

Author response: We would like to keep Table 1 in the main text. Figure 2 will be moved to supplements.

Figure 4: How much variation is there between altitude replicates? Maybe add a supplementary figure showing ellipsoids or individual sample points.

Author response: We agree with reviewer comment on Fig. 4. We attached new version of this figure and propose that it could be used in the main text instead of previous version. Please note that different length of the arrows (relative to centroid position) compared to previous version of this figure is due to different scaling.

[Figure]

Fig. 1 Comparison of mean daily $CO_2$ production at days 0-4, 4-12 and 13-14 (respiration presented in the manuscript).

[Figure]

Revised Fig. 4

---

## Author Response (AR1)

**Dear Associate Editor and Reviewers,**

please find below a detailed response to all Reviewers comments and questions regarding our manuscript. Since the Reviewers gave us a lot of useful hints and comments, we decided to completely rewrite the Discussion and Conclusions. We also significantly revised method, results (including figures) and supplementary materials to improve our manuscript. We hope that the changes we made will increase the quality of our manuscript in order to fulfill the requirements for publication in Biogeosciences.

Sincerely,

Petr Kotas and co-authors

**Reviewer 1**

**General comments**

The authors investigate the effect of horizontal (across a valley) and vertical (altitude) gradients on microbial community structure (PLFA), biomass and activity in High Arctic. They found that both gradient affect microbial parameters, with shift in the dominance of bacteria and fungi related to the chemistry of the bedrock. The study target interesting question, is relevant for publication in Biogeosciences and is overall well done. My main criticisms are the method used to measure microbial activity (main issue), and too many assumptions made outside the variables measured, going beyond what the results can show.

The measure of microbial activity seems unrealistic. First, 2 mm soil was used which was frozen and thaw prior to incubation in the lab. So, the microbial community and soil endure 1 freeze-thaw cycle + sieving that will affect OM availability and the microbial community. Then you left the samples for 14 days at 6 degrees before measuring the CO2 for 24h, which you define as "basal respiration". Fourteen days represent 1/4 of the summer (> 5 degrees) in the Arctic (low altitude) or even your entire summer for the high-altitude site (Table 1), this is a significant amount of time in the Artic. There is no justification and references used to explain why you made these choices. Overall, we can doubt that the high altitude produce high CO2 emissions in in-situ conditions and we can ask the values of your results regarding microbial activity. You did not discuss at any time the limitations of such measurement. We can imagine that the microbial community adapted better or took longer to adapt to incubation condition in high altitude soil explaining the higher CO2 emissions at 14 days. We can also imagine that at low altitude, because of the higher TOC, the CO2 emissions are high rapidly after thawing and after 14 there is not much activity, while for high altitude it took longer to mineralize more complex OM. In other words, your results of microbial activity could be just the results of your incubation/sample preparation. You need to fully acknowledge this in the article, and avoid any conclusion stating that high altitude is a hot spot of microbial activity because your data can't fully support this. You need to be much more conscious about the microbial activity result. Have you measure CO2 emissions over time?

Author response: We admit that it was inappropriate to call the measured respiration "basal". The characteristic we measured is rather the "potential respiratory activity". We agree with the reviewer that our methodological approach should be explained better. We also did measure the CO2 emissions during the incubation. We included the explanation in the Methods section (L130-140) and commented on this issue in the results (L238-244, Fig. 2) and discussed the results on L317-325.

The discussion and conclusions are too long and go far beyond what you can say based on your results. There are many sections you discuss about the dynamic of microbial community but you only did one sampling time. You can't make big conclusions about dynamic of the system, such as L362-376. You need to just briefly mention potential dynamic but don't go much further. Similarly, you

speak a lot about the effect of plant cover even you did not measure any parameters to characterize the plant cover (you also forgot to mention anything about mosses and lichens despite their importance in the Arctic) such as above ground biomass, root biomass, percentage cover (did you properly assessed it?), diversity. You just described the main vascular plants. So your section 4.4, is simply too long and not fully supported with your data. This entire section could be reduced in few sentences and focus on presence/absence of plants and not linked to microbial dynamic.

Author response: We agree with the reviewer's concern that discussion was too long and sometimes not organized and confusing. We completely rewrote the whole discussion. We also added information about soil crust, mosses and plant cover in the supplements and comment o it more specifically throughout the manuscript.

On the other hand, your discussion lack of putting your results in perspective with the literature, for example other studies investigating microbial community in bare/unvegetated soil (there is several article on this in the Arctic). It would be interesting to see if F/B ratio is similar in unvegetated a low altitude compared to other study.

Author response: We extensively searched the WOS again and included new sources in the manuscript. The discussion was completely rewritten.

Your main conclusion should be the absence of plant rather than altitude effect, especially if we consider that high altitude soil in the Arctic (especially High Arctic) is likely to be rare, and also extremely shallow (only few cm) and not a massive stock of C. So we can wonder of their importance?

Author response: We don't fully understand these remarks. Isn't the decreasing plant abundance due to the altitude effect? Please consider our sampling strategy (L 88-92). What connection is there with rareness of high altitude soils? We don't think that the high altitude soils doesn't deserve scientific interest (despite of their low C stocks and shallow soil profile), especially in context of proceeding global warming and future development of ecosystems in the Arctic. We would like to point out that the high elevation habitats form significant part of the non-glaciated landscape not only in Svalbard, but also in other parts of the northern circumpolar region. However, we completely revised our conclusions and discussion to make it more straightforward.

Also you should more conclude on the site effect you have (Gr1) as site is clearly an effect on microbial biomass (consistently lower) and structure for the high altitude site.

Author response: We agree with reviewer opinion. This issue was revised throughout the manuscript including result (section 3.3) and discussion.

Another point you miss in your discussion is the fact that the soil you study is always alkaline (are alkaline soils prevalent on Svalbard or acidic?). You need to discuss or conclude if your results would be similar on acidic soil in the Arctic and compare with the relevant literature as pH is a big driver of microbial community including in the Arctic. This is an important point to make and be more critical about your results.

Author response: We thank the reviewer for pointing out this problem. Soils on Svalbard are neutral or alkaline. The soil pH in other parts of the northern circumpolar region is more variable, ranging from acidic pH (mainly on granite and gneiss bedrock) to highly alkaline pH. We are aware of pH effect on microorganisms. This issue is discussed more thoroughly in the current manuscript version.

Finally, you find an effect of altitude on microbial biomass mainly when you divide the data by TOC. This bias the data and don't reveal hidden effect of altitude. Yes, you have less C at high altitude but you don't have more microbial biomass. The fact there is more biomass per unit of C is not of a major interest. Dividing your results by TOC is not important and bias your results. Focus on the altitude

effect on microbial community structure and ratio, and acknowledge that there is not a major effect of altitude on biomass but rather a site effect.

Author response: We agree with the reviewer that we should emphasize the spatial variability in the microbial biomass and activity data. The normalized data were removed from the manuscript.

**Specific comments**

Altitude and transect are both gradient and not only transect. This is really confusing in the text when you speak about gradient, as it is unclear if you speak about vertical or horizontal. You can't refer to the horizontal gradient as "gradient" and the vertical one as altitude. Decide if you speak about vertical or horizontal gradient, or altitude and transect. Change in the entire text, but don't go from "gradient" for horizontal and then use gradient also for "vertical". Be consistent.

Author response: We agree with the reviewer comment. We clarified this throughout the whole manuscript. We clearly distinguished between the effects of altitude (vertical aspect) and transect (horizontal aspect) in the revised manuscript.

**Introduction**

L20: true but it is simply related to less C from a plant origin, it does not mean that there is more biomass. This is not true as well for GR1. You are telling only one part of the story here.

Author response: Normalized microbial characteristics were removed from the manuscript.

L21: "the 2 dominant microbial groups" it sounds like it is unusual or a result on its own but in the same time with PLFA you have access to fungi and bacteria only. Just speak about fungi and bacteria.

Author response: We agree with this comment. Abstract was rewritten.

L23: you didn't measure microbial dynamic over time, only the change in soil temperature. So, I would focus on what you measured and not make assumption, especially in the abstract. Keep this for the discussion

Author response: Assumptions were removed from the abstract.

L25-26: the conclusion is an overstatement. In general, unvegetated area should be considered as previous studies showed. Speaking about high elevation as hotspots of microbial activity based on 1 measurement is an overstatement (see main comment).

Author response: We removed such conclusions from abstract.

L36-41: this is normal as high altitude usually have no soil present or are extremely shallow (few cm) and may not be as important in their distribution and volume than low altitude (see main comment).

Author response: We completely agree with the reviewer that the high altitude soils are more important in distribution and volume compared to high elevation soils. However, we wanted to point out in these lines that the soil microbial properties were not thoroughly studied yet along the altitudinal gradients in the Arctic. We consider the information given in these lines relevant.

L42: not true, you can assess the effect of changing microclimate not using altitudinal climate as you can be using different sampling time (which you did not do), different exposition, open top chambers etc. So, just focus on altitudinal but you can't say that other studies can't assess change of microclimate.

Author response: The sentence was removed.

L47: are the ranges of altitude comparable and are the ecosystems comparable?

Author response: No, they are not. We are writing here about general altitudinal trends. The referred studies investigated microbial communities in different latitudes and across different altitudinal ranges (please see our next response).

L51: you cite articles on complete different ecosystems, such as Fierer et al 2011 (tropical), Meng et al 2013 (forest)... Focus on the Arctic and no other biomes, and same ecosystems (i.e. tundra and not forest) as you will not expect to have the same trends. Clearly state the location and ecosystems the studies you cited are based on.

Author response: We are aware of that. General latitudinal, but also altitudinal trends in diversity of animals and plants are one of the most widely recognized patterns in ecology. However, they are not valid for the microbial diversity as we wanted to show here. The number of references about microbial diversity or community structure along altitudinal gradients from the Arctic is strongly limited.

L59: this is not true. Your study is also true at your sites and at other sites the effects will differ. The number of studies help us to determine the common drivers across different sites. Your study is not better than others at that level. For generalisation, you could have cited Chu et al 2010 as global study of microbial diversity across the Arctic (Soil bacterial diversity in the Arctic is not fundamentally different from that found in other biomes)

Author response: We completely agree with the reviewer opinion - our study is not better than others and the effect is site specific, which does not allow generalization - as we have written here. What is not true then? We didn't want to highlight our study here. However, we rewrote the sentence for clarity (L62-64).

L60: "Fundamental" this is a strong word, please rephrase.

Author response: Sentence was removed..

**Materials and Methods**

L80-84: any idea of the percentage plant cover? Did you measure it? You don't mention mosses and lichens, but they represent a large part of the plant cover in Arctic tundra and are completely missing from your description.

Author response: Regarding the primary producers, we have data about plant and lichenized soil crust percentage cover at the sampling sites. Mosses were relatively scarce at these locations. These data were added to the result section (L228-231, Fig. S5, Table S1) and discussed. However, we didn't assessed the plant diversity and biomass at the sampling sites.

L85: you speak about the bedrock in the entire article but you never define/describe it. Could you give some information on it.

Author response: We would like to thank the reviewer for pointing out this deficit. We have detailed information about geology of the Petunia Bay. The information was added to the site description (L83-86, see also L379-380).

L86: was an organic horizon present in the low altitude soils?

Author response: No, the soil profile is poorly developed. Based on our experience there is rather litter layer on the soil surface and then relatively homogeneous mineral soil layer overlaying coarse gravel.

L93: "kept frozen" at which temperature?

Author response: At -20 °C (L95)

L122: section 2.4 there are no references and no justification of your measurements choices: temperature, duration of incubation...?

Author response: The information was added (section 2.4) and the methodological approach was commented in results (L238-244, Fig. 2) and discussion (L317-325).

L131: why did you adjust the amount of soil based on TOC and in which way? Could this bring a bias if you have more soil for example in high altitude to compare you results?

Author response: We optimized the PLFA extraction protocol in our laboratory to suit wide range of different soil types (with respect to the amount of lipids and size of the SPE cartridge for lipid fractionation, analytical procedure and stock sample aliquots for eventual reanalysis). As we use 0.7-1g of soil with TOC content around 5% and the microbial biomass is usually proportional to TOC content, we adjust the sample size for C "poor" soils. This modification could not bias our results since the extraction efficiency is not affected and the PLFA yield is thus comparable for all samples. Rather the opposite is true – not accounting for low TOC content could lead to concentrations of particular PLFAs below the detection limit.

L154, 156: change "mL" to "ml"

Author response: The abbreviation mL was used in all recent articles published in BGS.

L161: I guess you checked also for homoscedasticity?

Author response: Yes, we checked the data also for homoscedasticity (L174).

L161: state clearly if you transformed or not the PLFA data.

Author response: The relative data were long-transformed (L174-175).

L163-164: the horizontal and vertical transect are both gradient (one vertical one horizontal). See comment at the beginning.

Author response: We agree with the reviewer. We clearly distinguished between the effects of altitude and transect in the revised manuscript.

L166: how was the forward selection done?

Author response: It was performed using CANOCO 5.0 software. The soil geochemical parameters were used as explanatory variables while the relative abundances of microbial groups (MCS) were used as the dependent variables (RDA). The test offer list of candidate variables sorted according to their contribution to total explained variation in the dependent variables, together with their significance. After selection of the best candidate, the contribution and significance of remaining candidates is recalculated to explain the remaining variability in the dependent data. Then the next candidate can be selected (of course only if significant).

L167: why did you use only P values adjusted by Holms corrections. Any reference for that?

Author response: We used the significant values adjustment to reflect the multiple tests performed on the same dataset. The Holm's correction follows the approach described *in* Holm (1979): A simple sequentially rejective multiple test procedure. Scand. J. Stat. **6**: 65-70. This procedure is slightly less conservative compared to the often recommended Bonferroni correction. On the other hand, it is a sequential procedure and takes into account that the candidate predictors with stronger effect were selected first. Thus it suits better for the

forward selection procedure (please see above our comment on the forward selection). Reference given in L183.

L172-173: it is really confusing when you speak about whole-plots vs splits-plots when you don't have a plot experiment. I am not sure what you refer to here.

Author response: As we mentioned in lines 185-187, we could assume that the characteristics of each sample will be auto-correlated with characteristics from other two samples taken from the same site (otherwise we had 9 independent transects and not three, which is not the case). In other words, the samples from the triplicate cannot be considered as independent samples due to relatively low inter-sample distance. The sampling design was in this context hierarchical with repeated measurements for each sampling site. We clarified this on L 185-187.

L176: what type of correlation did you use? Why there is no direct reference to correlation in the previous sentence?

Author response: We used the Pearson correlations to find out how tightly were two variables related to each other (L190-191).

**Results**

L190-192: this is a repetition of L 204. L192, it is also wrong what you say as the low altitude site show higher soil moisture than high altitude. Delete the sentence.

Author response: The results were largely rewritten and the duplications were removed (see sections 3.1 and 3.2).

L187, 203: which gradient are you talking about, be clear. L214: cite the Table you refer to L217-218: say if the correlation is positive or negative when you mention correlation even if it is given in brackets. L220: finish the sentence by "while increased in Gr2 and Gr3".

Author response: The changes were done according to reviewers comments throughout the manuscript.

L223-225: this should be given in the materials and methods and justify why you should use it. This problematic for me and can bias your results as mentioned in the main comment.

Author response: The normalization per TOC content was removed from the results.

L233: what is the "whole PLFA profile"? You did not use all the biomarkers in previous tests, it is not the same than MCS?

Author response: Comments on PLFA profile were removed.

L229-237: should cite figure 5? For example, L230 which figure you refer to?

Author response: There is no figure showing these results (L 249-251 in the current manuscript version). We think that figure showing the relation between selected environmental variables and MCS (Fig. 3 in the current manuscript version) are more important.

L250: change "typical" by "characterized"

Author response: Sentence was revised.

**Discussion**

L256: do you mean "did not" or "did" correspond. Looking at your plot, you have the same trend between soil and atmospheric, just few degrees' differences. Nothing surprising here. Your

explanation is not logical, snow will insulate the soil from air temperature, so having less snow should make the soil temperature more similar to air temperature but at the beginning you say they don't correspond. So, what are you trying to say?

L255-264: this whole section I am not sure what message you are trying to deliver. Nothing is really new here, and could be condensed in a shorter section.

L281: OM does not grow but increase. Change "growing" by "increasing" L285: "documented growing contribution of ", rephrase this is difficult English to understand

L288- 290: ok there is no vascular plant, but what about mosses and lichens? Could they be partly responsible for presence of sitosterol? You have to discuss about mosses and lichens in the article, you completely omitted to mentioned them, and I don't think they are not present on the soil. Lichens have a distribution up to high-altitude and you often find them on Svalbard on top of mountain even without any soil.

L306: this reference is a bit old, is there any more recent references done on a larger number of bacteria?

L307: what is the parental material?

L309-314: of course if you divide microbial biomass and activity by TOC you will find an altitude effect, because there is less plant input at higher altitude so lower TOC. It does not mean your microbial biomass is higher at high altitude or there is higher microbial activity. Is it really important or interesting to know that there is higher proportion of living microbial biomass per soil TOC content? Your site effect if stronger on microbial biomass (PLFA /g soil) than altitude (only present for GR1) which is an interesting result on its own and show the variability of the suppose vertical gradient.

L316-324: This is repetition of L309-324. You need to merge both section and make it shorter, and again that you have higher microbial biomass per TOC is not of a major interest.

L330-331: keep in mind that you work on alkaline soil and it might be difficult to compare to other soil which are acidic...L348: you are not working on "dynamics" because you only have one sampling time. Remove dynamic from the title as you can only make some assumption

L349: Why do you think it is low quality litter? Compare to what? The fact you have a high C/N does not mean low quality but just more OM and less degraded. Low OM quality usually have low C/N. This also could contradict what you say L354 "which released easily assimilable". Is low quality litter easily assimilable?

L352: So what? Ok it reduces the microbial biomass per unit of C, it just means that there is more C in the soil. You describe this as an issue but I don't see why it should be an issue? The microbial biomass does not decrease with increasing altitude! Sorry to repeat myself but you use the results you prefer to support your theory without considering your entire results.

L356-357: this is a strange wording which make the sentence difficult to understand. What do you mean by "inverse consequences from soil MCS compared to development of microbial communities"? In which way this is "inverse" and how do you have an inverse microbial community structure (or just different) and why you speak about development or young soil when you don't measure microbial growth or dynamic and the age of the soil?

L358: Are you talking of your study or not? You are not working on a "succession". You are again making too many assumption and extrapolation based on your results. This is also contradictory to what you say in the previous sentence. Here you say at the "maximal plant biomass" the fungi dominate, while you say bacteria dominate in the previous section and in general in the article.

L360: so, any conclusion? Can you really make this comparison based on 1 study? L362-376: your results do not support what you say. You have one sampling time point, don't make assumption on what you don't measure: dynamic. Focus on your results. You can say that the presence of ergosterol coincide with continuous dominance of fungi at high altitude sites, but you don't need an entire section about it. Delete most of this section into one or few sentences.

Author response: We agreed with most of the above-mentioned remarks. Based on these numerous comments, we decided to completely rewrite the whole discussion.

**Conclusion:**

L379: move "were" just before "characterized" L380: this is not true. Unless you divide by TOC, there is no consistent effect of altitude on biomass and activity. L381: can you really say that there is negligible effect of microclimatic conditions over the summer with only one date of sampling? Do you think you have enough resolution with your sampling strategy to assess the effect of summer

microclimate? L382: again, you use gradient without saying which gradient you are talking about and when you define in the material and methods "gradient" to refer to horizontal not vertical. L383: you need to clearly state the decrease in pH. The decrease is less than a pH unit and the soil remains slightly alkaline. This is important because your results are likely to be completely different on acidic soil. L384-385: again, what is the bedrock at your sampling site? L386-388: well, there is plenty of unvegetated area at low altitude and even when there are plants. Your thinking must be developed to unvegetated area not only at high altitude. Do plants will colonise high altitude soil which are only few cm thick with global warming? Also give a reference for the potential increase in plant cover in the Arctic as several articles were recently published. L389: you can't really say that it diminishes the variability because it depends on plant species colonizing new area, the bedrock as you say. You don't measure variability with PLFA, the resolution in the method you use is not high enough. L390: you don't measure microbial diversity, how do you know this could have a negative effect? L393: again not true, you don't have a considerable microbial biomass and your measure of microbial activity is questionable. You just can't make this conclusion L379-394: there is no mention of the site effect even if you clearly have a site effect on microbial biomass and activity. This should be clearly stated as the vertical gradient is directly affect by the horizontal one in relation (in your study) to bedrock.

Author response: The conclusions were completely revised.

**Reviewer 2**

**General comments:**

The study by Kotas et al. was focused on changes in microbial biomass, activity, and broad community structure (based on PFLA) along altitudinal gradients in the Artic. This question has great significance concerning the implications of global warming on these ecosystems. The study consists of 3 different transects represented by 4 different elevations, and for each sample the authors collected substantial amounts of data representing soil type, soil chemistry (pH, ion content and concentrations, TOC, TN, moisture content, and temperature ranges), and very briefly mention vegetation coverage. The authors try to disentangle the impacts of all these along with elevation on microbes using partial redundancy analysis as well as several other statistical approaches. They have a robust sample design with good replication to try and address this question.

I did have several issues with the manuscript. First, I found it very confusing that the authors kept referring to two different gradients, altitudinal (the main gradient of interest), and horizontal. However, this horizontal aspect is never discussed in the methods section and I assume it is referring to the south to north orientation of the 3 transects along the Petunia Bay. This needs to be clarified explicitly and its significance needs to be discussed. Is it expected there is a strong S-N effect? I assumed these 3 gradients were expected to be replicates of each other, but they have strong differences in soil characteristics and microbial community (particularly Gr1). This becomes more apparent in the Discussion, but the author's need to make this clear early on.

Author response: We agree with the reviewer opinion. We clarified this throughout the whole manuscript and clearly distinguished between the effects of altitude (vertical aspect) and transect (horizontal aspect). The 3 transects were expected to be replicates of each other. We didn't expect any variability in soil geochemical or microbial characteristics which could be ascribed to the differences in orientation of the selected transects. Opposite was true - we did our best to select similarly oriented transects (slopes on the western coast of Petunia Bay) in order to minimize the effect of distinct slope orientation.

I also had concerns with their microbial respiration data and the authors need to justify their choice of a 2 week pre-incubation at 6 C. The pre-incubation will burn off all the labile carbon and drastically alters this respiration rate. This needs discussed as it can substantially alter the conclusions of a large portion of the paper.

Author response: We agree with the reviewer that we have to justify and discuss our methodological approach. The methodology was chosen according to available knowledge and our experiences with similar experiments. We insist that the presented respiration data corresponds to in situ microbial activity. We measured the CO2 emissions during the incubation and included these data in the manuscript. We also included justification of our methodological choices in the Methods section (L130-140) and commented on this in result section (L238-244, Fig. 2) and discussed the results on L317-325.

The discussion is too long and wordy. I found it difficult to understand the main points the authors were trying to convey. It seemed to be rushed relative to the excellent writing of the rest of the manuscript and has multiple grammar issues. I also think that there was too much superfluous material that distracts from the main message. The authors spend a great deal of time discussing impacts due to plant biomass, but have no data presented quantitatively examining plant communities, biomass, root biomass, etc. A lot of this can be safely removed, especially in sections 4.1 and 4.4, as the degree of detail discussed doesn't add too much to the broader implications of the study.

Author response: We agree with these comments. Discussion was revised, shortened, and previous sections were merged into two main parts. We also provided the information about plant and lichenized soil crust percentage cover as these data are recently available for the sampling sites (please see L228-231, Fig. S5, Table S1).

With some mostly editorial changes focusing on clarifying the findings I think this paper represents a significant contribution towards Arctic research and understanding the environmental parameters shaping microbial communities in this sensitive area.

**Specific Comments**

L124: I was interested in why the authors decided to pre-incubate the soils at 6 C (far above the mean of -3.8C, and below the max of 16.2, as well as different from the 5 C cut-off used in L186])?

Author response: The incubation temperature of 6 °C was chosen as it represents mean summer soil temperature along the whole elevational gradient (mean summer temperatures for particular elevational levels ranged from 5.3 to 7.1, see Table 1; the mean summer temperature across the whole gradient is 6 °C). Justification is given in section 2.4.

Also, why did the authors choose to pre-incubate for 2 weeks at this temperature? Is this typical for these kinds of measurements? I would think you want to minimize the pre-incubation time to prevent a strong bottle effect, as well as removing all your labile carbon.

Author response: Our methodological choices and more detailed description of our incubation experiment are given in section 2.4. We further commented implications of our measurement in L238-245 and L317-326.

L126: Is the specific respiration ratio typical to compare with the field? Is it possible to convert PLFA to a more generalizable unit (such as per cell, per g biomass etc.) using conversion factors?

Author response: We excluded the specific respiration rate from results. However, we don't think that conversion of soil PLFA content to microbial biomass carbon (or per cell) could add any value. The conversion factors vary in the literature sources and are inevitably affected by cell morphology (membrane area versus cell biovolume). There is different PLFA to microbial biomass ratio not only for fungi and bacteria, but also for bacterial cells differing in size and shape. As the fungi to bacteria ratios varied significantly between sites, we consider any recalculation using a single conversion factor as speculative and hardly employable for comparison with other studies based on measurements of soil microbial carbon content (e.g. by chloroform fumigation method).

L144: Is there a reference to support this sum? Are you not overcounting the bacterial contribution by summing general bacterial biomarkers with specific bacterial group biomarkers (Actinos, G-, G+)? Would it not be preferable to us general fungal : general bacterial only?

Author response: The bacterial abundance is in majority (if not all) of papers using PLFA as quantitative measure of microbial biomass calculated as a sum of all markers specific to bacteria. The specific bacterial groups (Actinobacteria, G-, G+) belongs to bacteria and they need to be considered when calculating the F/B ratio. Considering only general bacterial markers, which are specific to bacteria but cannot be ascribed to one of the above mentioned bacterial groups, would lead to significant overestimation of fungal presence in the soil (references e.g. Frostegård and Bååth 1996, Biol. Fert. Soils 22, 59-65; Bååth and Anderson 2003 SBB 35, 955-965; Kaiser et al. 2010, New Phytologist 187, 843-858).

L189: Maybe change "In contrary" to "In contrast".

Author response: Sentence was rewritten.

L214: Maybe add at the end "and was instead transect specific". I realize this is implied, but I feel it makes it clearer.

Author response: Sentence was rewritten.

L213 - L227: This section is confusing to me. It is very surprising that microbial activity (as you assayed it) is not related to carbon or nitrogen content and is instead related to positively with Ca and negatively with Mg. I worry the trend in increasing respiration with altitude is due to the pre-incubation.

Author response: This relationship between respiration and base cation availabilities was surprising also for us. However, the microbial activity (respiration in this case) doesn't have to correspond with biomass as was shown previously (Šantrůčková and Straškraba, 1991, SBB 23, 525-532). Based on the background data from our respiration measurements (please see above our response to general comments), we insist that the presented respiration data are not a result of our pre-incubation step and can be used as potential respiratory activity of soil microbes. We thus believe that soil geochemical properties such as high magnesium availability can be very important drivers of microbial activity and abundance in these arctic soils. Moreover, the studies of Webb (Webb, 1949; reference in the manuscript) support the assumption, that parent material with very high Mg2+ content could have such negative effect on microbes.

L228: Write out "Microbial Community Structure" in the header of this section.

**Author response: Done**

L229: Gradient here is the transect? Does this mean there is a continuous change along the S-N transects or that each is different?

Author response: Yes, gradient is transect here. The results mean that there is a significant shift in the MCS not only between elevations, but also significant differences between transects in horizontal direction. The use of horizontal (transect) and vertical (altitude) aspects was emphasized throughout the manuscript.

L230: Nice to see so much explained due to altitude!

L231: Which gradients? Elevation or between the transects? Please fix or clarify this terminology!

**Author response: Terminology was clarified throughout the manuscript.**

L229 – L233: These few sentences are quite confusing and I think readers would be helped if you clarify. If I understand, the microbial community structure is impacted by elevation, but even more so by how the soils change with elevation? You ran multiple different tests to parse out these effects at different levels? Also, is microbial community structure here a relative score or absolute values?

Author response: We will clarify these statements. Let us to offer brief explanation: the microbial community structure significantly changed along the elevational gradients and between transects (ie. both factors, transect and elevation, were significant). The significant effects of transect and elevation can be well explained by spatial variability in the soil geochemical properties which were determined (ie. horizontal and altitudinal variability in the soil properties). The MCS used here and in general throughout the manuscript are relative abundances of microbial groups (not scores, see L175-176 in Method section).

L237: Re-running the analysis with the selected variables was non-significant? Can you clarify this statement? Why do you want to run the forward selection if the variables selected do not significantly explain the microbial community composition? Is the main message of this part, that these variables are not significant while altitude is?

Author response: These results were removed from the manuscript.

L240 – L251: Nice results! I think this is more interesting that the previous paragraph. However, there are a lot of grammar mistakes here, some listed below. Maybe re-write this section for clarity.

Author response: Section was rewritten and clarified.

L243: missing a space L247: "A similarly significant trend" L248: Change to PFLAs. L249: change discrepant to disparate

249: change discrepant to disparate

Author response: Was corrected.

L249: Consider re-writing, this is a very long sentence that can be shortened, maybe "The most disparate site in terms of MCS was the highest elevation sampled along Gr1. It was typified by a high abundance of PLFAs specific to Actinobacteria and a lower abundance of fungal PFLAs compared to analogous sites along Gr2 and Gr3."

Author response: Sentence was rewritten.

L255: What does this sentence mean?

Author response: The whole section was shortened and clarified.

L265: "positive surface energy balance had a strong.."

Author response: Corrected

L273: This is an incredibly important but difficult to decipher sentence. I think a lot of the sentences above it can be shortened or removed, but this should be clarified. Do you mean that "Mean temperatures and temperature stability did not change with altitude in this study"? [Therefore, variations in your parameters due to altitude are not simply due to temperature differences?] Here I would start off with a stronger statement of what you mean, and then offer your support.

Author response: We mean that mean temperatures and temperature stability (diurnal temperature fluctuation) does not change with elevation as we expected – ie. temperature will decrease with increasing elevation and the microclimate will be less stable in higher altitudes. We also expected generally higher fluctuation of soil moisture. However, we found very similar temperature conditions in the lowest and highest elevations, while the mid-elevated sites experienced warmer but less stable summer soil microclimate. The most important microclimatic parameter thus seemed to be the length of vegetation season and its effect on vegetation. The whole section was rewritten and clarified.

L277: Extremely important to clarify what gradient you are talking about here.

Author response: Clarified.

L277: Are you missing a "not". This is a confusing sentence.

**Author response: Discussion was completely rewritten.**

L281 – L296: Simplify this! It is too wordy and difficult to follow. E.G. "We explain this discrepancy by the proximity of glacier stream, which could wash away the upper soil organic layer during abnormal spring-melt events in the past", can be changed to "The only exception was the lowest site of Gr2 which had similar OM content to higher elevation sites along the other transects. This is likely due to the proximity of a glacier stream, which would wash away the topsoil during a flood."

Author response: We agree that the paragraph is too wordy. Paragraph was completely revised.

L284: "vascular plants also influenced"

L286: Please provide a citation for this.

Author response: citation provided (L337).

L288 – L290: Is this important for your findings?

Author response: Rewritten

L290: Lots of grammar issues.

Author response: Rewritten

L292: Or high lichen components at high elevation?

Author response: We agree that the importance of lichens must be thoroughly discussed. However, lichens contain algal and cyanobacterial photobionts so there is not a conflict with our statement.

L298 – L314: You need to discuss the implications of your pre-incubation step in this section. It can also be clarified or simplified for the readers.

Author response: We discussed the implications of our pre-incubation step. The whole paragraph was revised (see our comments to incubation experiment above).

L304-L308: Please include relevant concentrations of the Mg inhibitory effect here.

Author response: We would like to thank the reviewer for this comment. The inhibitory concentrations of  $Mg^{2+}$  in solution were above 5 p.p.m and 50 p.p.m. for G- and G+ bacterial species, respectively (Webb 1949, Microbiology 3, 410–424). The limiting concentrations will be mentioned in the discussion.

L309 – L314: This is a nice summary. However, the normalized characteristics are inherently dependent on the soil OM, so isn't their increase directly due to the OM decrease?

Author response: The normalized microbial characteristics were removed from the manuscript.

L323 – L324: Please clarify this statement. What shift in resources lead to the slow accumulation of low quality OM? What are the ramifications of your pre-incubation when you are suggesting some samples are enriched in more recalcitrant OM?

Author response: Rewritten

L327 – L336: A lot of speculation. Is all this necessary

L337 – L347: Very speculative.

Author response: We believe that  $Mg^{2+}$  availability is very important factor shaping MCS along the transects. It largely explained the trends in G-/G+ bacteria ratios (compare Table 3 and Fig. 6c, d in the manuscript). It was shown that growth of G- and G+ bacteria is limited at very different  $Mg^{2+}$  concentration levels (difference of one order of magnitude, see our response to comments on L304-308). The  $Mg^{2+}$  availability in the investigated soils exceeded these limiting concentrations, especially for G- bacteria (considering all available  $Mg^{2+}$  in soil solution and average soil moisture content 30%, the  $Mg^{2+}$  concentrations ranged approximately from 50-420 p.p.m.). We thus consider the given interpretation of observed shifts in MCS due to  $Mg^{2+}$  availability ( $Mg^{2+}$  availability was retained by RDA with forward selection of explanatory variables) as critical evaluation of relevant literature. However, we admit that statements about substitution of fungi by Actinobacteria are speculative and will be removed. The section was completely revised.

L384: "bedrock chemistry were recognized as the main factors"

Author response: Rewritten

L387 – L388: A confusing sentence, consider revising.

Author response: Rewritten

Figure2: Consider moving either this figure, or Table1 to the supplemental information to shorten the main paper.

Author response: We would like to keep Table 1 in the main text. Figure 2 was moved to supplements.

Figure 4: How much variation is there between altitude replicates? Maybe add a supplementary figure showing ellipsoids or individual sample points.

Author response: We agree with reviewer comment on Fig. 4. New version of the figure showing the variability between altitude replicates and transects (Fig. 3 in the current version of our manuscript).

[revised manuscript text omitted]

Summer | M eans
Winter | Means
Year | M in daily means
Winter | M ax daily means
Summer | M ean daily
amplitude
Summer | M ax daily
amplitude
Summer | Number of days
with daily mean >
0 °C | Number of days
with daily mean >
5 °C | Positive soil
surface energy
balance |
|-----|---------------------|------------------|------------------|---------------|----------------------------|----------------------------|------------------------------------|-----------------------------------|---------------------------------------------|---------------------------------------------|--------------------------------------------|
|     | 25                  | 5.8              | -3.6             | -0.8          | -7.0                       | 11.2                       | 5.2                                | 10.9                              | 110                                         | 62                                          | 615                                        |
|     | 280                 | 7.1              | -5.7             | -2.7          | -10.3                      | 14.5                       | 8.5                                | 18.2                              | 96                                          | 54                                          | 571                                        |
|     | 520                 | 5.8              | -8.9             | -4.9          | -15.8                      | 14.7                       | 8.1                                | 17.7                              | 91                                          | 40                                          | 480                                        |
| 0 _ | 765                 | 5.3              | -9.5             | -6.6          | -17.1                      | 11.6                       | 5.5                                | 14.0                              | 51                                          | 11                                          | 290                                        |

Table 2. Geochemical characteristics of soils along the studied altitudinal transects (Tr1-Tr3). Means  $\pm$  SD (n = 3) are given in the

upper part of the table. Results of two-way ANOVAs (F-values) of the effects of transect (Tr), altitude (Alt) and their interaction (Tr x Alt) are presented in the lower part of the table.

|   | 1.5. 1    | 3.              |                                  |                                 |                                  | G 2+                      | 2+                                  |                                | +                                    |
|----------|-----------|-----------------|----------------------------------|---------------------------------|----------------------------------|--------------------------------------|-------------------------------------|--------------------------------|--------------------------------------|
| transect | altitude  | soutype         | soil moisture                    | рн                              | CEC                              | Ca                                   | Mg 2                     | K                              | Na                                   |
|          | [m a.s.l] |                 | [%]                              |                                 | [meq/100g -1 ]        | [mg g -1 ]                | [mg g -1 ]               | [µg g -1 ]          | [µg g -1 ]                |
| Tr1      | 25        | sandy loam      | $a_{28.4 \pm 2.5}$               | $b_{7.8 \pm 0.1}$               | $a_{35.8 \pm 0.4}$               | $\mathbf{b}_{4.9\pm0.2}$             | $\mathbf{c}_{0.50\pm0.03}$          | $b_{104 \pm 2.3}$              | $a_{16.0 \pm 1.4}$                   |
|          | 275       | sandy loam-loam | $\mathbf{b}_{18.0\pm0.5}$        | $^{\boldsymbol{b}}_{7.9\pm0.2}$ | $^{\mathbf{b}}27.4 \pm 2.3$      | $\mathbf{b}_{5.2\pm0.6}$             | $^{\mathbf{c}}_{0.55\pm0.08}$       | $\mathbf{b}_{81\pm8.8}$        | $^{\textbf{bc}}_{8.4\pm1.3}$         |
|          | 525       | loam            | ${}^{\mathbf{b}}_{18.6 \pm 2.5}$ | $\mathbf{b}_{8.1\pm0.1}$        | $\mathbf{b}_{30.3\pm0.7}$        | ${}^{\boldsymbol{b}}_{4.3\pm0.4}$    | $\mathbf{b}_{0.85\pm0.04}$          | $a_{160 \pm 18.1}$             | b
11.3 ± 1.1               |
|          | 765       | clay-loam       | $^{\mathbf{c}}_{12.1\pm1.8}$     | $\mathbf{a}_{9\pm0.0}$          | $\mathbf{b_{26.8\pm 2.3}}$       | $^{\mathbf{a}}_{19.8\pm1.0}$         | $^{\mathbf{a}}_{1.25\pm0.06}$       | $c_{11 \pm 2.7}$               | $^{\mathbf{c}}_{7.3\pm0.0}$          |
| Tr2      | 25        | sandy loam      | a
21.1 ± 2.4           | $c_{7.8 \pm 0.1}$               | $b_{25.6 \pm 2.7}$               | b $_{14.7 \pm 2.6}$           | $c_{0.19 \pm 0.01}$                 | $ab_{52 \pm 4.0}$              | a 13.2 ± 1.7                  |
|          | 275       | sandy loam-loam | a 21.1 ± 2.4              | $c_{7.9 \pm 0.1}$               | $\mathbf{b}_{30.3 \pm 1.7}$      | $\mathbf{ab}_{16.5\pm1.1}$           | $\mathbf{b}_{0.26\pm0.01}$          | $a_{59 \pm 4.3}$               | $\substack{\textbf{ab}}{10.1\pm1.7}$ |
|          | 525       | sandy loam-loam | a 21.7 ± 5.3              | $^{\boldsymbol{b}}_{8.4\pm0.1}$ | $\boldsymbol{b}_{30.8\pm1.1}$    | $\mathbf{c}_{7.8 \pm 1.6}$           | $^{\mathbf{a}}_{0.34\pm0.01}$       | $a_{69 \pm 3.3}$               | $\mathbf{ab}_{9.6 \pm 1.8}$          |
|          | 765       | loam            | $a_{22.5 \pm 1.7}$               | $^{\mathbf{a}}_{8.8\pm0.1}$     | a $45.1 \pm 0.5$      | a 27.9 ± 9.3                         | ${}^{\boldsymbol{b}}_{0.25\pm0.01}$ | $\mathbf{b}_{41\pm8.8}$        | $\mathbf{b}_{8.1\pm1.4}$             |
| Tr3      | 25        | sandy loam      | a 39.5 ± 1.4              | $\boldsymbol{b}_{8.1\pm0.1}$    | $a_{49.4 \pm 2.1}$               | $c_{7.7 \pm 0.3}$                    | $a_{0.20 \pm 0.03}$                 | b $_{52\pm5.3}$         | a 17.1 ± 1.1                  |
|          | 275       | sandy loam-loam | $^{\textbf{ab}}_{31.9\pm2.9}$    | $\boldsymbol{b}_{8.1\pm0.1}$    | $\mathbf{b}_{39.2\pm5.4}$        | $^{\boldsymbol{b}}10.8\pm0.6$        | $^{\mathbf{a}}_{0.21\pm0.01}$       | $\mathbf{ab}_{59\pm1.9}$       | $^{\mathbf{a}}_{18.5\pm0.5}$         |
|          | 525       | loam            | $ab_{28.2 \pm 6.5}$              | $\mathbf{b}_{8\pm0.1}$          | $\mathbf{b}_{34.9\pm3.0}$        | $\substack{\textbf{ab}\\13.0\pm4.6}$ | $\mathbf{a}_{0.22\pm0.00}$          | $a_{66 \pm 6.6}$               | $\mathbf{a}_{18.4\pm3.1}$            |
|          | 765       | loam            | ${}^{\mathbf{b}}_{22.5 \pm 1.7}$ | $\mathbf{a}_{8.8\pm0.1}$        | ${}^{\mathbf{b}}_{30.6 \pm 3.9}$ | $a_{14.2 \pm 0.1}$                   | $\mathbf{b}_{0.16\pm0.00}$          | ${}^{\mathbf{b}}_{52 \pm 1.6}$ | ${}^{\mathbf{b}}_{9.9 \pm 0.2}$      |
|          |           | d.f.            |                                  |                                 |                                  |                                      |                                     |                                |                                      |
| Tr       |           | 2               | 31.4 ***                         | 0.10                            | 22.1 ***                         | 6.43 **                              | 634 ***                             | 51.7 ***                       | 36.2 ***                             |
| Alt      |           | 3               | 11.1 ***                         | 98 ***                          | 4.61 *                           | 14.1 ***                             | 66.9 ***                            | 74.9 ***                       | 18.7 ***                             |
| Tr x Alt |           | 6               | 5.07 **                          | 5.6 ***                         | 20.5 ***                         | 0.83                                 | 60.6 ***                            | 31.6 ***                       | 3.94 **                              |

Different letters indicate significant differences between sampling sites along particular transects (P < 0.05; upper part of the table). Statistically significant differences are indicated by: \* P < 0.05, \*\* P < 0.01, \*\*\* P < 0.001 (lower part of the table).

Table 3. Total soil carbon (TOC) and nitrogen (TN) contents, their molar ratios, contents of sitosterol in TOC and sitosterol / brassicasterol ratios and soil PLFA contents in soils along the altitudinal transects (Tr1-Tr3). Means  $\pm$  SD (n = 3) are given in the

brassicasterol ratios and soil PLFA contents in soils along the altitudinal transects (Tr1-Tr3). Means  $\pm$  SD (n = 3) are given in the upper part of the table. Results of two-way ANOVAs (F-values) of the effects of transect (Tr), altitude (Alt) and their interaction (Tr x Alt) are presented in the lower part of the table.

| transect | altitude   | TOC                        | TN                          | TOC/TN                     | Sitosterol                | Sitosterol / Brassicasterol |  |
|----------|------------|----------------------------|-----------------------------|----------------------------|---------------------------|-----------------------------|--|
|          | [m a.s.l.] | [mg g -1 ]      | [mg g -1 ]       |                            | [µg g -1 TOC]  |                             |  |
| Tr1      | 25         | ° 70.6 ± 13.4              | b 5.0 ± 1.01         | b $12.1 \pm 0.2$    | с 534 ± 62.8   | b 5.5 ± 0.4          |  |
|          | 275        | b $_{21.1 \pm 1.9}$ | $a_{2.0\pm0.29}$            | ab $9.0 \pm 0.7$    | bc 521 ± 140       | b $5.3 \pm 0.8$      |  |
|          | 525        | b $_{18.5 \pm 4.2}$ | $a_{1.8 \pm 0.31}$          | $ab_{8.8 \pm 0.7}$         | ab 293 ± 66.5      | ${}^{\bf b}_{4.7\pm1.0}$    |  |
|          | 765        | a 4.4 ± 1.5         | $\mathbf{a}_{0.5 \pm 0.07}$ | a 7.9 ± 2.6         | a 81.1 ± 2.7       | a $2.3 \pm 0.4$      |  |
| Tr2      | 25         | ab $30.6 \pm 4.8$   | a $1.9 \pm 0.40$     | $c_{13.7 \pm 0.9}$         | bc $515 \pm 44.9$  | b $6.7 \pm 0.7$      |  |
|          | 275        | b 37.2 ± 5.0        | a 3.0 ± 0.26         | b $10.7 \pm 0.7$    | $c_{616 \pm 143}$         | b 5.6 ± 1.2          |  |
|          | 525        | a 24.4 ± 7.8        | a $1.9 \pm 0.64$     | b $_{9.8 \pm 1.2}$  | ab 299 ± 73.3      | a $2.9 \pm 0.4$  |  |
|          | 765        | a 21.6 ± 3.6        | $a_{2.8 \pm 0.20}$          | $a_{6.7 \pm 0.6}$          | a 161 ± 36.9       | a 2.7 ± 0.7          |  |
| Tr3      | 25         | $c_{81.1 \pm 8.7}$         | b $6.1 \pm 0.38$     | b $11.5 \pm 0.7$    | b $_{587 \pm 144}$ | b $6.4 \pm 2.1$      |  |
|          | 275        | b 62.2 ± 9.1        | $ab_{4.8 \pm 0.32}$         | b $11 \pm 0.7$      | ab 370 ± 42.9      | $a_{4.2 \pm 0.7}$           |  |
|          | 525        | ab 39.6 ± 11.4      | $\mathbf{a}$ 4.8 ± 0.32     | b $_{10.6 \pm 0.6}$ | a 270 ± 112        | a 3.3 ± 1.0          |  |
|          | 765        | a 23.1 ± 3.9        | $a_{2.5 \pm 0.37}$          | a 7.9 ± 0.2         | a 151 ± 37.8       | a 3.1 ± 0.9          |  |
|          | d.f.       |                            |                             |                            |                           |                             |  |
| Tr       | 2          | 27.8 ***                   | 31.5 ***                    | 1.57                       | 0.79                      | 1.04                        |  |
| Alt      | 3          | 42.4 ***                   | 26.4 ***                    | 23.6 ***                   | 28.4 ***                  | 14.4 ***                    |  |
| Tr x Alt | 6          | 8.33 ***                   | 11.3 ***                    | 1.96                       | 1.34                      | 2.17                        |  |

769Different letters indicate significant differences between sampling sites along particular transects (P

---

## Referee Report (RR1)

General Comments:

In this edited manuscript, the authors do an excellent job stating their results and placing them within the current body of knowledge, without over interpreting them. I felt that the authors responded to all valid comments I had previously made and corrected me on comments I had made that were not valid.

I was particularly impressed with their discussion on the microbial respiration rates and appreciate that the authors included data for multiple measurements throughout the thawing process. They also did an excellent job distinguishing between the three transects and the altitudinal effects which were difficult to understand in the previous version. The rewritten discussion was much clearer and easier to digest, and I thought the main points were well supported and well stated. Finally, the authors did a good job at speculating about impacts of climate change within the constraints of their results and the conclusions section did an excellent job summing up the results of this paper.

One issue that needs to be taken care of is the grammar throughout the manuscript. There are too many grammatical errors to count and I've highlighted a few examples below. I suggest it be thoroughly edited by an English speaker.

Specific Comments:

L240: Should reference Fig 2b, c.

L249: Seems strange to start this section off with the interactive effect, when the RDA showed that most of the variation could be constrained to an altitudinal effect, while only 8% was explained by the interaction term (if I'm reading your results correctly).

L252: Doesn't this indicate that ALL the effects (altitude, transect, altitude:transect) were largely driven by the soil variability rather than just the interactive term? I agree that the interaction term is interesting and it is nice you can relate it to Mg++, however, Figure3 shows all 3 transects oriented in the same ordination space from the lower left to the upper right alongside increasing elevation. It seems like the text is focusing so strongly on the interaction, when the two standalone variables have more explanatory power. I do want to note that I think the discussion section does a very nice job interpreting these results.

Table1: Please rename first column to Site Altitude [m a.s.l.]

Figure 2b: Should the Y-axis title be "Potential Respiration"?

Figure 2: This is completely a personal preference on my part, but I like the horizontal ticks you have in part c between the sites. You may want to consider adding the longer ticks in parts (a) and (b). This is not completely necessary though, since you nicely change the color of the bars for each transect.

L798: "Results of RDA", is this supposed to be the figure title? It is not a complete sentence and is strange in the middle of the legend.

Also, check this figure legend in general for grammar errors. "The correlation between the abundance of the main microbial groups (bold italic) and the soil geochemical parameters that were retained by forward selection of all the explanatory variables collected. The altitude of sampling sites was used as a

supplementary variable…. Dotted lines indicate environmental variables retained by the forward selection model."

Figure 4a. Here I recommend adding the larger x-axis tick marks as seen in 4b, since there is no color change distinguishing between the transects.

Grammar examples:

L20: Grammar error, "we did not observe…".

L22: "Mainly due to differences in bacterial PLFA compositions, but also systemic altitudinal shifts in MCS related to…"

L32: "fundamental roles…"

L37: "Offer a great opportunity"

L38: "presence of vegetation, and…"

L65: "…we conducted a study aiming to assess…."

L229: "while mosses covered a very small proportion…"

L242-L245: The grammar issues in this sentence need to be addressed as it is difficult to understand.

---

## Author Response (AR2)

**Author comments**

*Reviewer 1*

General Comments:

In this edited manuscript, the authors do an excellent job stating their results and placing them within the current body of knowledge, without over interpreting them. I felt that the authors responded to all valid comments I had previously made and corrected me on comments I had made that were not valid. I was particularly impressed with their discussion on the microbial respiration rates and appreciate that the authors included data for multiple measurements throughout the thawing process. They also did an excellent job distinguishing between the three transects and the altitudinal effects which were difficult to understand in the previous version. The rewritten discussion was much clearer and easier to digest, and I thought the main points were well supported and well stated. Finally, the authors did a good job at speculating about impacts of climate change within the constraints of their results and the conclusions section did an excellent job summing up the results of this paper. One issue that needs to be taken care of is the grammar throughout the manuscript. There are too many grammatical errors to count and I've highlighted a few examples below. I suggest it be thoroughly edited by an English speaker.

> We thank the Reviewer for generally positive evaluation of manuscript revision and valuable comments. To improve the language quality and avoid grammatical errors, we used professional services for language proofreading.

Specific Comments:

L240: Should reference Fig 2b, c.

> Done. We thank the reviewer for this notice.

L249: Seems strange to start this section off with the interactive effect, when the RDA showed that most of the variation could be constrained to an altitudinal effect, while only 8% was explained by the interaction term (if I'm reading your results correctly).

> We agree. The first two sentences were reworded and the comment on interaction term was moved to the second sentence (L251-252).

L252: Doesn't this indicate that ALL the effects (altitude, transect, altitude:transect) were largely driven by the soil variability rather than just the interactive term? I agree that the interaction term is
interesting and it is nice you can relate it to Mg++, however, Figure3 shows all 3 transects oriented in
the same ordination space from the lower left to the upper right alongside increasing elevation. It seems like the text is focusing so strongly on the interaction, when the two standalone variables have more explanatory power. I do want to note that I think the discussion section does a very nice job interpreting these results.

> You are right. We wanted to say this on L252. To make it more clear, we added "separate and interactive effect of altitude and transect" to the sentence on L 253.

Table1: Please rename first column to Site Altitude [m a.s.l.]

Done

Figure 2b: Should the Y-axis title be "Potential Respiration"?

Yes. We changed the title.

Figure 2: This is completely a personal preference on my part, but I like the horizontal ticks you have in part c between the sites. You may want to consider adding the longer ticks in parts (a) and (b). This is not completely necessary though, since you nicely change the color of the bars for each transect.

The figure was edited as recommended.

L798: "Results of RDA", is this supposed to be the figure title? It is not a complete sentence and is
strange in the middle of the legend.

Sentence was removed from the legend.

Also, check this figure legend in general for grammar errors. "The correlation between the abundance of the main microbial groups (bold italic) and the soil geochemical parameters that were retained by forward selection of all the explanatory variables collected. The altitude of sampling sites was used as a supplementary variable....Dotted lines indicate environmental variables retained by the forward selection model."

We revised the legend.

Figure 4a. Here I recommend adding the larger x-axis tick marks as seen in 4b, since there is no color
change distinguishing between the transects.

Done

Grammar examples:
L20: Grammar error, "we did not observe…".

Corrected.

L22: "Mainly due to differences in bacterial PLFA compositions, but also systemic altitudinal shifts in MCS related to…"

Revised

L32: "fundamental roles…"

Corrected

L37: "Offer a great opportunity"

Corrected.

L38: "presence of vegetation, and…"

> Corrected.

L65: "…we conducted a study aiming to assess…."

> Corrected.

L229: "while mosses covered a very small proportion…"

> Corrected.

L242-L245: The grammar issues in this sentence need to be addressed as it is difficult to understand

> The sentence was reworded and clarified (L 242-246).

*Reviewer 2*

Main comments:

The authors did a good job to respond to all the comments from both reviewers. They changed the figures to clarified their results, and rewritten the discussion delivering a clearer message. I only have few specific corrections and two main point that the authors need to address below.

> We thank the Reviewer for generally positive evaluation of manuscript revision and valuable comments. We tried to address all the remaining reviewers concerns as follows.

Thanks for clarifying the methods used for measuring soil respiration and also to show the respiration at day 4 as well as day 13. An important point that need to be added, it is not only a question of the effect of freeze-thaw cycle, temperature and length of the incubation on the samples, but you apparently also sieved the sampled at 2 mm before freezing the samples (L94-95). Sieving the sample will have stronger effect than one cycle of freeze-thaw, releasing organic matter, breaking down soil crust and aggregates and also exposing microorganisms to different level of O2. Coupled with the temperature and length of incubation, this could drastically alter the trends in your results and stimulate activity in some samples more than others. Together, and as you mentioned, you measured potential respiration, but you need to clearly say that these conditions are not in situ, and a potential effect of the incubation procedure (sieving, temperature, thawing) cannot be discarded. You have to mention the effect of sieving (L318-319) which could have a stronger effect on soil biological crust than other soil, and have to conclude that you can't discard a method effect (L324-326). I stress the fact that this is a requirement to clearly state the potential limitation of your measurement.

> We agree with the reviewer that sieving is an important treatment prior to respiration measurements, which can influence the respiration rates. In accord, we added comments on this issue in L301-307.

Section 4.1 is still too long and mainly describe the site and present results instead of discussing them. The section brings little information. For example, the first sentence is similar than the material and methods, just a site description; L277, this result is already given L199 in the results section. And this is true for most of the section. Unless, you have

novel results compared to the literature, or those results can directly explain the microbial results, this section is not useful and could be reduced in few sentences or deleted. This is especially true from L272-285. The sentence L285-287 is unclear and should be rephrased (if kept) and related to the microbial community. Line 287-290 is also detached from the microbial data and miss linking those results together.

> Even though the microclimatic measurements could not be directly linked to the presented microbial characteristics, we considered these data as important and valuable to portrait the soil conditions along the elevational transects. However, we agree with the reviewer that the section was too long and the given information was in some cases mentioned twice throughout the manuscript. We thus significantly shortened the first paragraph of the discussion (L272-284).

The next section (L291-300) is not at his place in section 4.1. This is (Mg) partly discussed L366-380. I would merge both sections and avoid repetition.

> We reduced the paragraph to avoid repetitions and merged the information into the last paragraph of our discussion (L 356-365).

Overall, I don't think you need section 4.1 and 4.2 but just one discussion nicely split in different section without headers.

> In accord with reviewer recommendation, the current discussion is a continuous text without headers.

Specific comments:

L12: delete "proceeding"

> Done

L18: change in the whole text "basal respiration" to potential respiration"

> Done

L20: change "in" by "on"

> Done

L32: "play a fundamental"

> Done

L39: delete "The proceeding"

> Done

L52, 55, 56, 58: this was already in my initial review (reviewer 1); you need to clearly state which ecosystems the articles you site work on. As you said in your reply, the altitude trend for microorganisms does not work like for plants and animals. So, where the study took place is likely to have a strong effect on the results. You have to state which ecosystems the study you site work on, this will help the reader to have a better understanding. As it read, the

reader could think that all these studies took place in the Arctic but it is not the case. Please, give the ecosystems. This is also true for the discussion.

Done (L52-62)

L65: delete "alpine"

Done

L65: change "the arctic alpine" to "arctic"

Done

L65 "we conducted a study"

Done

L142: it is difficult to believe there is no significant difference between day 4 and day 13 when figure 2 b and c shows that at day 4 (Fig 2c) potential respiration is around 2 times higher than at day 13 (Fig 2b). Unless what you want to say is that you have the same difference between the altitude regardless of the measurement date? Be more clear.
Corrected

It is also not true that you only present day 13 as suggested at the end of Line 142, as you present day 4 and 13. Please correct the sentence.

Sentence deleted

L144: change "defined" by "determined"

Done

L186-187: it is not true that you can't consider the triplicate as independent. Microbial ecology shows that you can have more similarity in samples taken km away from each other than few cm away. This is especially true for your design when you sample from vegetated area to bare soil, while you clearly show in your study that bare soil are different. So, you could consider your triplicate as independent. This is a general comment, as I am not asking to change the statistics here.

We partly agree with this reviewer comment on soil microbial characteristics. However, with respect to geochemical properties of soil and underlying bedrock, which will be perhaps more similar in the few meter range compared with km away, we decided to evaluate the data as we described. Nevertheless, the effect of such choice on the output statistics is very low and rather decreases the statistical significance of performed tests.

L221: change "Oppositely", by "In contrast"

Done

Section 3.3: it is worth mentioning that the daily rate is around 2 times higher at day 4 than day 3. Would be also interesting to mention how it compare with day 12. Even if this is not

the main message you want to deliver, it is an interesting result which could be briefly presented.

Done (L240-245)

L233: "was significantly positively correlated"

Done

L243: "change "had" by "at"

Done

L258: change "Oppositely" by "In contrast"

Done

L260: delete "an"

Done

L265: change to "The soil with the poorest TOC and richest Mg concentration at the highest site on Tr1..."

Corrected

L272: "characterised by a four"

Corrected

L294: delete "In result,"

Done

L316, 327, 328, 329, 330: give the ecosystems the studies are working on

Locations given according to the reviewer recommendation

L320: give the days of incubation in brackets with what you define as flush, adaptation, stabilization

Done (L308)

L323: change "accord" by "agreement"

Done

L330: "the majority… was associated with…"

Done

L354: change "at more elevated sited" by "with altitude"

Done

L355: change "typical" by "characterised"

Done

L355: "unfavourable" depends of the microorganisms as you mention, what is unfavourable for one is favourable for the other. It is better to characterise the conditions rather than saying favourable or not.

Done (L343)

L357-358: you need to acknowledge here that you did not measure the F/B ratio on the incubated samples use to measure soil respiration. Thus, you can't be sure that the fungi explain such results. You could have a complete shift of your community as stated L327.

Done (L351-352)

L358: change "prosper" by "grow"

Done

L361: what do you mean by "benign". Don't use such word but rather define the type of soil/conditions

Done (L350)

L 364: change "likely" by "could"

Done

L370: "Tr1 had higher actinobacteria and phototrophic microorganisms abundance"

Sentence reworded

L390: delete or rephrase, what does "uniform" mean in that context? Do not use "concurrently"

Sentences were reworded

Figure 2: b and c are both potential respiration, it would make more sense to give the incubation day (4, 13) which is more clear, especially in the caption and y axis. "flush respiration" and "potential respiration" are not self-explanatory in the figure. The caption should not need the support of the text to be understood.

The Figure 2 was edited as recommended